# Syntax vs. Semantics: How Transformers Learn Deep Dependencies

**Jiangrui Zhao** [1]    **Xiaoting Du** [1] [†]

## Abstract

Large Language Models demonstrate remarkable syntactic fluency, yet the optimization dynamics governing their acquisition of deep semantic dependencies remain poorly understood. We propose a mechanistic framework that models this learning process as a competition between Surface Statistics and Deep Semantics. Our theoretical analysis identifies a "Gradient Starvation" phenomenon where the error signals for sparse semantic dependencies are actively suppressed during early optimization. This suppression impedes the learning of structural reasoning and causes its emergence to manifest as a sudden phase transition. Furthermore, this framework offers a mechanistic basis for the effectiveness of Chain-of-Thought (CoT) strategies. By externalizing intermediate reasoning steps into concrete tokens, CoT effectively bypasses the suppression regime inherent to implicit reasoning. We validate these findings across scales ranging from toy transformers to production models (Llama-3.1-8B, Qwen2.5-Coder-7B). Finally, guided by this theory, we propose a topology-aligned contrastive objective that explicitly rectifies the gradient geometry. Experiments on variable binding tasks demonstrate that our method achieves an improvement that is over 2× larger than that obtained via standard cross-entropy fine-tuning. Code will be publicly available at: https://github.com/jr-zhao/Deep-Dependencies/tree/main.

## 1. Introduction

The hallmark of general intelligence is not merely surface-level fluency, but the ability to acquire and exploit *Deep Dependencies* that extend beyond local patterns. Such

[1] School of Computer Science (National Pilot Software Engineering School), Beijing University of Posts and Telecommunications, Beijing, China. [†]Corresponding author . Correspondence to: Xiaoting Du <duxiaoting@bupt.edu.cn>.

*Proceedings of the $43^{rd}$ International Conference on Machine Learning*, Seoul, South Korea. PMLR 306, 2026. Copyright 2026 by the author(s).

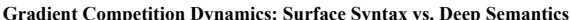

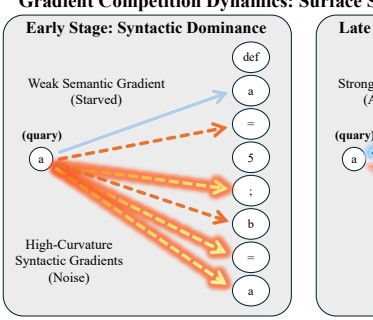 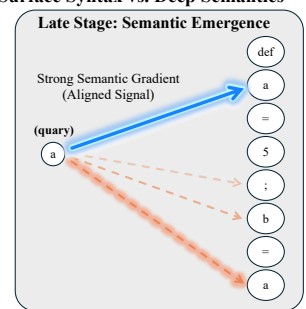

*Figure 1.* **Gradient Competition Dynamics**. Left: High-curvature syntax (orange) starves the semantic signal. Right: The model aligns with the deep topological dependency (blue).

dependencies require models to recover latent relational structure from context, a capability long argued to be central to systematic generalization (Fodor & Pylyshyn, 1988; Marcus, 1998). While recent work has demonstrated that Transformer-based models can eventually capture these abstract dependencies (Vaswani et al., 2017; Olsson et al., 2022), the optimization dynamics governing their emergence remain poorly understood. In practice, models reliably master local, high-frequency structure early in training, yet only later acquire deep dependency reasoning, often via an abrupt qualitative transition rather than gradual improvement (Wei et al., 2022a; Power et al., 2022; Davies et al., 2023; Wu et al., 2025).

We argue that this phenomenon is not architectural but dynamical. The operators required for deep dependency reasoning are structurally attainable, existing as latent linear subspaces within standard Transformer geometry (Dhayalkar, 2025; Boix-Adsera et al., 2023). However, accessing these subspaces during optimization is non-trivial. We propose a mechanistic framework that characterizes training as a competition between **Surface Statistics** (syntax) and **Deep Semantics** (latent dependency structure). During early optimization, high-curvature gradients induced by frequent local patterns dominate the loss landscape, a phenomenon known as "Gradient Starvation" (Pezeshki et al., 2021). These gradients suppress weaker, low-curvature signals associated with long-range dependencies, effectively masking semantic learning until a critical phase transition occurs.

The tension between syntactic structure and semantic con-

tent has been widely studied, either through architectures that explicitly separate the two (Russin et al., 2019; Felhi et al., 2022; Caucheteux et al., 2021), the injection of structural priors such as dependency trees (Bai et al., 2021; Gong et al., 2022; Zhou et al., 2020; Kalyanpur et al., 2020), or probing analyses that question whether syntactic competence entails semantic generalization (Weissweiler et al., 2022; Ahuja et al., 2025; Alleman et al., 2021). However, these approaches largely treat syntax and semantics as static representational properties or architectural design choices. In contrast, our work models them as competitors in the gradient landscape, providing a mechanistic interpretation for why syntactic structure tends to be learned earlier and how it may interfere with the acquisition of deep dependencies during training.

This framework yields two central predictions. First, it explains the hierarchical emergence of reasoning: optimization pressure forces attention heads to prioritize recovery of coarse relational topology before resolving fine-grained values, leading to a characteristic staged development of dependency circuits. Second, it provides a causal account of the effectiveness of Chain-of-Thought (CoT) prompting (Wei et al., 2022b). By externalizing intermediate states, CoT alters the learning geometry by injecting additional gradient pathways, partially bypassing the starvation regime that constrains implicit reasoning.

To empirically validate these dynamics, we adopt a multi-stage experimental strategy. Because natural language lacks unambiguous ground truth for latent dependency structure, we primarily use source code and Abstract Syntax Trees (ASTs) as a controlled proxy for semantic topology (Allamanis et al., 2017; Chen, 2021). We trace the emergence of dependency circuits from controlled setting toy models to intermediate checkpoints of Pythia (Biderman et al., 2023b), and finally demonstrate the effectiveness of our topology-aligned intervention on production-scale models, including Qwen2.5-Coder-7B (Hui et al., 2024; Yang et al., 2024) and Llama-3.1-8B (Dubey et al., 2024).

# 2. Mechanistic Characterization: The Dynamics of Semantic Emergence

In this section, we provide a theoretical framework for the emergence of Semantic Binding Circuits (or Contextual Dependency Circuits). We formulate the problem as the competition between Surface Statistics (Syntax) and Deep Semantics (latent dependency structure). We demonstrate that this competition is governed by a signal-to-noise ratio (alignment ratio) dynamic, where high-curvature syntactic features initially dominate the optimization dynamics through gradient starvation, suppressing effective updates along semantic directions and delaying the emergence of systematic reasoning until a critical phase transition occurs.

## 2.1. Problem Setting: Surface Structure vs. Deep Binding

Let sequence $\mathcal{S} = \{x_1, \ldots, x_T\}$ be the input. We distinguish two functional dependencies: **Syntax** ($\mathcal{F}_{syn}$), comprising local, high-frequency patterns (e.g., separators); and **Semantics** ($\mathcal{F}_{sem}$), comprising long-range dependencies that recover the computational topology (specifically, mapping a Query token to its informational predecessor in the program graph).

## 2.2. The Geometry of Competition: Gradient Starvation

**Assumption 2.1** (Spectral Disparity). We assume optimization pressure is uneven. The syntactic subspace $\mathcal{U}_{syn}$ spans eigenvectors with large curvature $\lambda_{syn}$, while the semantic subspace $\mathcal{U}_{sem}$ has smaller curvature $\lambda_{sem} \ll \lambda_{syn}$. The initial residual $r_0$ has non-trivial energy in $\mathcal{U}_{syn}$.

This assumption is empirically verified in App. A.1.

**Proposition 2.2** (Gradient Starvation via Spectral Bias). *Consider the local quadratic approximation of the loss* $\mathcal{L}(\theta) \approx \mathcal{L}(\theta^*) + \frac{1}{2}(\theta - \theta^*)^T H(\theta - \theta^*)$. *Let* $e = \theta - \theta^*$ *denote the parameter residual. The gradient* $g \approx He$ *can be decomposed in the Hessian eigenbasis* $\{(\lambda_k, v_k)\}$ *as:* $g = \sum_k \lambda_k(v_k^T e)v_k$, *where scalar* $(v_k^T e)$ *is the projection of the residual onto the $k$-th eigenvector. Assuming* $\lambda_{syn} \gg \lambda_{sem}$ *and that the residual has comparable average projection magnitudes on both subspaces, the gradient magnitude is dominated by syntactic components:*

$$\frac{\|g_{syn}\|_2}{\|g_{sem}\|_2} \approx \frac{\lambda_{syn}}{\lambda_{sem}} \cdot \sqrt{\frac{m_{syn}}{m_{sem}}} \gg 1, \qquad (1)$$

*where $m_{syn}$ and $m_{sem}$ are the effective dimensions of the two subspaces. Consequently, the update direction $-g$ is dominated by the syntactic component, and the optimization makes much slower progress along semantic directions, yielding a masking (gradient-starvation) effect for $\mathcal{F}_{sem}$ (Proof: App. C.1).*

## 2.3. Suppression Mechanism: Softmax Saturation

We analyze how early syntactic convergence impedes semantic learning. Let $s_{sem}$ be the logit for the correct semantic token, and $A_{syn} \approx 1$ be the attention weight on a distracting syntactic token.

**Lemma 2.3** (Gradient Suppression Factor). *Let $s_{sem}$ be the attention score for the correct semantic token and $A_{sem}$ be its softmax weight. In the saturated regime dominated by syntax ($A_{syn} \approx 1$), we have $A_{sem} \approx 0$. The gradient of the loss $\mathcal{L}$ with respect to the score $s_{sem}$ is given by:*

$$\frac{\partial \mathcal{L}}{\partial s_{sem}} = \sum_k \frac{\partial \mathcal{L}}{\partial A_k} \frac{\partial A_k}{\partial s_{sem}}$$
$$= A_{sem} \left( \frac{\partial \mathcal{L}}{\partial A_{sem}} - \sum_k A_k \frac{\partial \mathcal{L}}{\partial A_k} \right). \qquad (2)$$

*Thus, the magnitude scales linearly with the attention weight:* $\left|\frac{\partial \mathcal{L}}{\partial s_{sem}}\right| \propto A_{sem}$.

**Theorem 2.4** (The Vanishing Gradient Barrier). *Under the top-1 domination regime where a single syntactic score $s_{syn}$ dominates the softmax normalizer, we have $A_{sem} \approx \exp(-(s_{syn} - s_{sem}))$. When the model is in the "Syntactic Phase" ($A_{syn} \to 1$), the effective learning signal for the semantic component vanishes exponentially with the score difference $s_{syn} - s_{sem}$. Under bounded $\frac{\partial \mathcal{L}}{\partial A_k}$, we have*

$$\left|\frac{\partial \mathcal{L}}{\partial s_{sem}}\right| = O\left(e^{-(s_{syn} - s_{sem})}\right) \to 0 \quad \textit{(Derivation:AppendixC.2)}.$$

*Implication: This multiplicative suppression creates a Gradient Barrier. Even if the definition token contains useful information, the signal is attenuated. Semantic learning is thus effectively stalled until the syntactic confidence $A_{syn}$ degrades (e.g., via conflicting patterns or regularization), raising $A_{sem}$ above a learnable threshold.*

## 2.4. Emergence via Subspace Alignment

Once the attention head escapes saturation, the interaction matrix $W_{QK} = W_Q^T W_K$ aligns with the underlying semantic topology.

**Assumption 2.5** (Error-Noise Orthogonality). We decompose each token embedding as $x = \mu + \delta$, where $\mu$ lies in a low-dimensional signal subspace and $\delta$ is zero-mean context noise.

**Assumption 2.6** (Signal Decomposition). We additionally assume $\mathbb{E}[\delta] = 0$ and $\mathbb{E}[\delta_j \delta_i^T] \approx 0$ for unrelated token pairs. Let $\gamma_{ji} = \frac{\partial \mathcal{L}}{\partial s_{ji}}$ be the scalar error signal flowing back to the attention score. We assume that under the data distribution $\mathcal{D}$, the backward error signal $\gamma$ is uncorrelated with the forward context noise $\delta$ inherent in the embeddings:

$$\mathbb{E}_{\mathcal{D}}[\gamma_{ji} \cdot \delta_i] \approx 0.$$

This assumption is empirically verified in App. A.3.

**Proposition 2.7** (Hebbian Reconstruction of Dependency Topology). *The attention score is the bilinear form $s_{ji} = x_j^T W_{QK} x_i$. The gradient of the loss with respect to the interaction matrix $W_{QK}$ accumulates as the outer product of inputs:*

$$\nabla_{W_{QK}} \mathcal{L} = \sum_{j,i} \gamma_{ji} x_j x_i^T, \quad \text{where } \gamma_{ji} = \frac{\partial \mathcal{L}}{\partial s_{ji}}.$$

*Substituting the signal decomposition $x = \mu + \delta$ (Assumption 2.5) and invoking the orthogonality assumption(justification in App. C.3), the expected update direction becomes:*

$$\mathbb{E}_{\mathcal{D}}[\nabla_{W_{QK}} \mathcal{L}] \approx \mathbb{E}[\gamma_{ji}] \cdot (\mu_{use} \mu_{src}^T). \quad (3)$$

This confirms that the optimizer naturally pushes $W_{QK}$ to align with the rank-1 outer product $\mu_{use} \mu_{src}^T$, which represents the topological edge in the computation graph.

## 2.5. Phase Transitions and the Critical Threshold

To quantify the transition, we define the *Alignment Ratio* $\rho(t)$. We define the ideal topological operator $M_{sem}$ as the rank-1 outer product encoding the dependency edge:

$$M_{sem} = \mu_{use} \mu_{src}^T.$$

Note that $M_{sem}$ is generally not symmetric (as usage $\neq$ source). We define $\rho(t)$ as the squared cosine similarity between the current gradient update and this target operator:

$$\rho(t) = \frac{\langle \nabla_{W_{QK}} \mathcal{L}, M_{sem} \rangle_F^2}{\|\nabla_{W_{QK}} \mathcal{L}\|_F^2 \cdot \|M_{sem}\|_F^2 + \epsilon}. \quad (4)$$

This metric $\rho(t) \in [0, 1]$ measures geometrical alignment(threshold derivation: App. C.4). Based on these thresholds, we identify three distinct phases:

**Phase I: Syntactic Dominance** ($\rho(t) < \tau_{low}$). Due to the high curvature of syntactic features (Prop. 2.2), gradients are dominated by the syntactic subspace, leading the model to preferentially fit local n-gram patterns. Under this regime, the vanishing-gradient barrier induced by softmax saturation (Thm. 2.4) suppresses effective updates along semantic directions, preventing the stable formation of binding circuits.

**Phase II: The Crossover** ($\rho(t) \in [\tau_{low}, \tau_{stable})$). This is the volatile heuristic phase. As syntactic loss saturates, the update direction begins to acquire a non-trivial component along the semantic operator $M_{sem}$, causing $\rho(t)$ to rise from near-zero to an intermediate regime. However, the alignment is not yet stable: competing heuristics and residual syntactic correlations induce high-variance gradients, preventing consistent consolidation.

**Phase III: Systematic Emergence** ($\rho(t) \geq \tau_{stable}$). When $\rho(t)$ exceeds a stability threshold $\tau_{stable}$, the gradient becomes aligned with the topological operator $\mu_{use} \mu_{src}^T$. The interaction matrix $W_{QK}$ locks onto this low-rank(nearly 1) direction, effectively implementing the copy mechanism(Empirical verification of rank collapse: App. A.2).

**Corollary 2.8** (Sparsity via Competitive Alignment). *The emergence of binding circuits is constrained by the residual-stream additivity across heads. Let $e := \nabla_x \mathcal{L}$ denote the backpropagated error at the residual stream of the layer. The gradient magnitude for a specific head $h$ is controlled by the projection of this error onto the semantic subspace:*

$$\|\nabla_{W_{QK}^{(h)}} \mathcal{L}\| \lesssim C \cdot \|Proj_{\mathcal{U}_{sem}}(e)\|. \quad (5)$$

*Due to initialization variance, a early-aligned head $h^*$ crosses $\tau_{stable}$ first, effectively reducing $\|Proj_{\mathcal{U}_{sem}}(e)\|$. While the semantic residual does not vanish, it decreases sufficiently such that for the remaining heads $h \neq h^*$, the alignment signal is overwhelmed by the high-curvature syntactic noise (Prop. 2.2). Consequently, the effective SNR for lagging heads drops below the critical threshold required to escape the syntactic regime, confining the semantic function to a sparse subset of heads.*

*Remark* 2.9 (Prescriptive Implication). This framework suggests that to accelerate the emergence of reasoning (Phase III), one can explicitly manipulate $\rho(t)$. Strategies such as syntax damping (masking loss on syntactic tokens) or logic densification (removing redundant syntactic markers from data) effectively boost $\rho(t)$, shortening the stagnation of Phases I and II.

This sharp transition in generalization coincides with what has been described in prior work as "grokking". In our framework, grokking can be interpreted as the moment when the alignment ratio $\rho(t)$ crosses the stability threshold $\tau_{stable}$, marking a qualitative reorganization of gradient geometry rather than gradual performance accumulation.

*Remark* 2.10 (Theoretical Prediction: Priority of Pointer Passing). Our framework implies a strictly ordered emergence of compositional circuits. Consider a dependency chain $i \xrightarrow{\text{ref}} j \xrightarrow{\text{val}} k$ (e.g., $i$="capital", $j$="France", $k$="Paris"). The model must decide whether to attend $i \to j$ (Pointer Passing) or $i \to k$ (Direct Value Fetching).

**Mechanism: The Precedence of Topological Neighbors.**
We predict that the alignment ratio $\rho(t)$ for the pointer edge $(i \to j)$ consistently crosses the stability threshold $\tau_{stable}$ before the value edge $(i \to k)$. The edge $i \to j$ typically represents a Type-Schema relationship (e.g., Operator $\to$ Operand, or Attribute $\to$ Entity), which corresponds to a dense, lower-curvature region of the semantic manifold compared to the sparse(Visualized via subspace probing in App. A.4), high-frequency map to the specific value $k$. Mathematically, the projection of the gradient onto the pointer operator $M_{ptr} = \mu_i \mu_j^T$ dominates the projection onto the value operator $M_{val} = \mu_i \mu_k^T$:

$$\langle \nabla \mathcal{L}, M_{ptr} \rangle_F \gg \langle \nabla \mathcal{L}, M_{val} \rangle_F \implies \rho_{i \to j}(t) > \rho_{i \to k}(t). \tag{6}$$

Thus, the dynamics force the attention head to first lock onto the intermediate node $j$, establishing the "Pointer Passing" circuit as the primary topological relaxation (SNR analysis: App. C.5).

**Subsequent Integration via Residual Streams.** Once the connection $i \to j$ is established, the residual stream property facilitates the secondary emergence of direct fetching. The update $x_i \leftarrow x_i + W_{OV} x_j$ physically moves the repre-

sentation of the pointer $j$ into the position $i$. For a deeper layer $L$, the distance to the value $k$ is effectively shortened. The attention mechanism can now utilize the copied content of $j$ to resolve the dependency $j \to k$ directly from position $i$, appearing phenomenologically as a "skip-connection" to the value $k$.

**2.6. Accelerating Semantic Emergence via CoT**

We distinguish the mechanistic role of CoT in two regimes: ① **Inference (State Externalization):** Implicit reasoning requires resolving multi-hop dependencies ($x \to \cdots \to y$) in a single pass. CoT externalizes intermediate states, factorizing the difficult conditional $P(y|x)$ into local steps $P(z|x)P(y|x, z)$, reducing the complexity of attention retrieval. ② **Training (Gradient Injection):** Supervision on traces introduces a local objective $\mathcal{L}_{total} = \mathcal{L}_{y|z} + \mathcal{L}_z$.

**Proposition 2.11** (Parallel Gradient Pathway). *In implicit learning, the gradient $\nabla_\theta \mathcal{L}_{imp}$ must backpropagate through the entire downstream computation of $y$. This signal is vulnerable to attenuation if the downstream dependency is weak or noisy. CoT supervision injects a local gradient term $\nabla_\theta \mathcal{L}_z$. Crucially, this term is independent of downstream correctness (i.e., it does not depend on the gradient flow from $y$). This short-circuits the long credit assignment chain, ensuring that the binding circuit receives a valid learning signal even when the implicit downstream signal is vanishing or confused.*

*Remark* 2.12 (Why CoT Helps Reasoning). We define Relative Gain as the ratio of semantic updates with vs. without CoT. For deep compositional tasks where the implicit signal is starved due to vanishing downstream gradients ($\|P_{sem}(\nabla^{imp})\| \to 0$), the injected gradient dominates, effectively initiating the alignment process (Gain $\gg$ 1)(Jacobian analysis: App. C.6). For shallow tasks where implicit signals are sufficient, the gain is marginal (Gain $\approx$ 1).

**3. Controlled Experiments: Verifying the Signal-to-Noise Dynamics**

We utilize a small Transformer trained on synthetic code as a controlled setting environment. While our theoretical framework (Section 2) applies to general sequential reasoning, code allows us to precisely decouple Surface Statistics (syntax tokens like =, ; ) from Deep Semantics (variable dependencies), a separation often obscured in natural language. All detailed experiment settings can be seen in App. B.1.

The training data is constrained to (i) variable assignment (binding definition) and (ii) arithmetic operations (binding usage). We design three experiments to empirically verify the theoretical predictions: specifically, the existence of gradient starvation (Prop. 2.2), the release from Softmax

Saturation (Thm 2.4), and the causal effect of manipulating the alignment ratio $\rho(t)$ (Remark 2.9).

## 3.1. Visualizing Gradient Starvation

We first track the training dynamics to validate the gradient starvation hypothesis (Theorem 2.2). We monitor both the loss curve and the gradient norms associated with syntactic tokens ($\mathcal{T}_{syn}$) versus variable tokens ($\mathcal{T}_{sem}$).

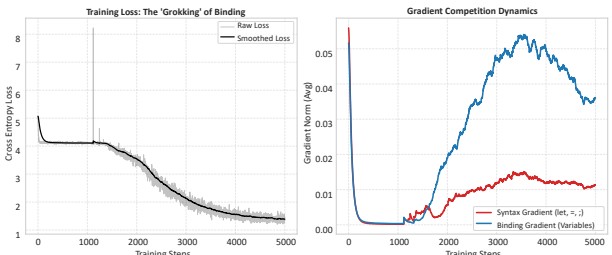

*Figure 2.* Left: training loss over optimization steps. Right: average embedding gradient norms for different token groups, where the red curve corresponds to syntactic tokens (e.g., LET, =, ;) and the blue curve corresponds to variable tokens (e.g., v0, v1, ...).

As shown in Figure 2, the model exhibits a clear two-stage learning process. In the early phase, despite large semantic errors reflected by a high loss, gradients on variable tokens (blue curve) remain suppressed and comparable to the syntactic baseline, consistent with gradient starvation: under Softmax Saturation (Section 2.3), semantic errors fail to propagate to the embeddings.

Once syntactic gradients decay and attention patterns destabilize, a sharp phase transition occurs. The sudden surge in variable gradients signals the release of the Softmax gate, enabling the accumulated semantic error to update the identifier subspace. This delayed gradient burst indicates active inhibition of the semantic circuit rather than its absence.

## 3.2. Escaping Softmax Saturation

We examine the mechanism of the phase transition by tracking attention allocation. We employ a curriculum setup: training on variable assignment until convergence, then introducing addition tasks at step 1500.

Figure 3 illustrates the behavior predicted by Theorem 2.4. Early in training, attention is locked onto syntactic anchors (red dotted line), enforcing $A_{sem} \approx 0$ and suppressing semantic error propagation as predicted by Eq. 2.3, resulting in low task accuracy.

As syntactic error diminishes, attention confidence weakens and a crossover occurs: attention to operand tokens (blue line) exceeds syntactic attention, unlocking the gradient gate and enabling semantic alignment in the identifier subspace. This transition coincides with a rapid accuracy increase and

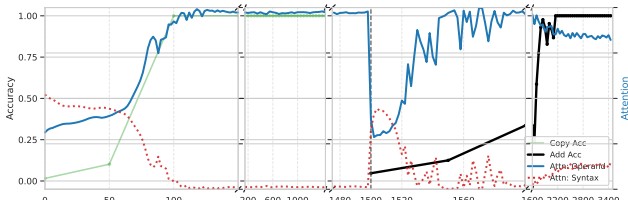

*Figure 3.* Task accuracy and attention distributions during curriculum training. The green line shows accuracy on variable assignment(copy), and the black line shows accuracy on addition after the task switch. The blue line denotes attention to operand tokens (e.g., variables involved in the calculation), while the red dotted line denotes attention to syntactic tokens (e.g., LET, =, ;).

is consistently observed in both variable assignment and addition tasks as the alignment ratio $\rho(t)$ crosses the critical threshold $\tau_{stable} \in (0,1)$.

## 3.3. Causal Validation of alignment ratio Thresholds

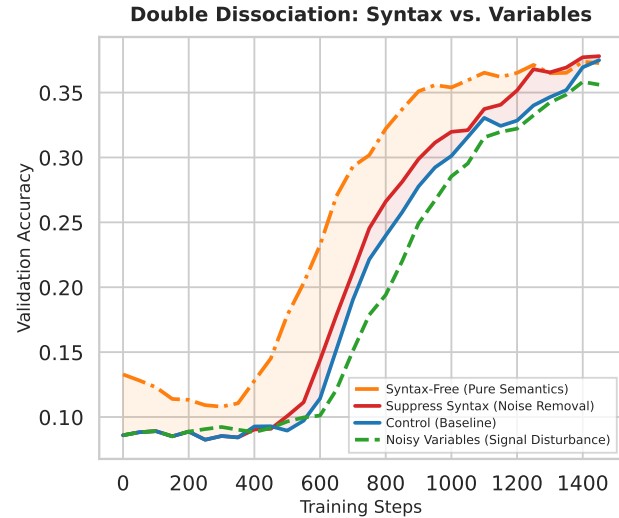

*Figure 4.* Accuracy comparison across three conditions: Suppress Syntax (red solid line), Control (blue solid line), and Noisy Variables (green dashed line).

Finally, we perform targeted interventions to further investigate the role of the alignment ratio $\rho(t)$. Based on the prescriptive implication in Remark 2.9, we compare a standard Control model against three variants: (1) a syntax-suppressed model, where gradients on syntactic tokens are artificially masked (equivalent to $\nabla_{syn} \to 0$); (2) a variable-disturbed model, where noise is injected into variable-token gradients; and (3) a syntax-free setting, where high-frequency syntactic markers are removed while preserving the underlying semantic dependency structure.

The results in Figure 4 provide strong empirical support for the gradient-starvation hypothesis. When gradients associated with syntactic tokens are suppressed (red), the up-

date direction becomes less dominated by non-semantic components, which increases the alignment ratio $\rho(t)$ and accelerates alignment with semantic dependency structure. Similarly, the syntax-free setting (orange) exits the early optimization plateau substantially earlier, suggesting that syntactic competition itself contributes to delayed dependency learning. In contrast, injecting noise into variable-token gradients (green) perturbs the semantic alignment direction, reduces the correlation between $\nabla_{W_{QK}}\mathcal{L}$ and $M_{sem}$, lowers $\rho(t)$, and consequently delays semantic emergence.

We additionally replicate the same transition behavior in a controlled natural-language entity-binding task, where removing high-frequency functional tokens similarly accelerates semantic alignment. This suggests that the proposed mechanism is not limited to code syntax, but may reflect a broader optimization phenomenon in deep dependency learning.

This double dissociation provides strong evidence that the emergence of binding is closely linked to the competition ratio $\rho(t)$. It suggests that the syntactic suppression strategy proposed in our theory is a viable method for accelerating reasoning emergence in larger models.

### 3.4. Summary of Findings

These experiments confirm the three pillars of our theoretical framework: (1) Gradient Starvation is a major contributing factor behind the initial learning plateau; (2) **Softmax Saturation** acts as the gatekeeper mechanism; (3) The emergence of reasoning is a phase transition governed by the semantic alignment ratio $\rho(t)$.

Crucially, the success of the Syntax Suppression intervention demonstrates that the learning of structure and reasoning are competitive processes, suggesting that data curation strategies for LLMs should prioritize maximizing logic density over syntactic perfection.

## 4. Mechanistic Experiments: The Dynamics of Emergence and Composition

Having established theoretical bounds in a controlled setting, we extend our analysis to the training dynamics of real-world LLMs using Pythia (Biderman et al., 2023b;a; van der Wal et al., 2025), a family of decoder-only transformers trained on the Pile (Gao et al., 2020; Biderman et al., 2022). Pythia provides dense intermediate checkpoints, enabling a longitudinal analysis of semantic circuit emergence. Leveraging these checkpoints, we validate our framework across three mechanistic scales: (i) alignment phase transitions ($\rho(t)$) in Pythia-160M to test gradient starvation, (ii) the structural priority of pointer formation in Pythia-1.4B, and (iii) signal propagation in compositional chains, illustrating how CoT bypasses the gradient bottle-

neck. All detailed experiment settings can be seen in App. B.2

### 4.1. Dynamics of the Alignment Ratio $\rho(t)$

To validate the subspace competition hypothesis, we track the geometric alignment $\rho(t)$ in Pythia-160m. Since the Transformer architecture parameterizes queries and keys separately, we cannot directly observe the gradient of the composite interaction matrix $\nabla_{W_{QK}}\mathcal{L}$. Instead, we compute $\rho(t)$ using the effective gradient reconstructed via the product rule: $\nabla_{\text{eff}} \approx (\nabla_{W_Q}\mathcal{L})^T W_K + W_Q^T(\nabla_{W_K}\mathcal{L})$. This metric captures the implicit update to the attention geometry, ensuring mathematical equivalence to our theoretical definition while respecting the model's factored parameterization.

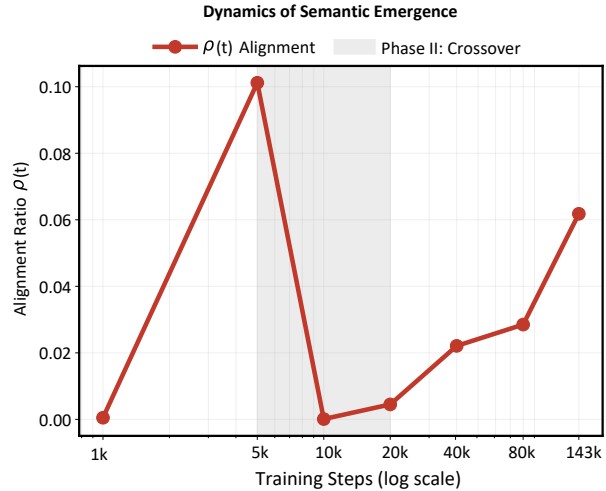

*Figure 5.* The evolution of alignment ratio $\rho(t)$ in Pythia-160m reveals a transition from syntactic gradient starvation to Systematic Emergence, punctuated by a volatile crossover phase (shaded) that reflects subspace competition.

Figure 5 reveals a distinct non-monotonic transition. Initially, $\rho(t) \approx 0$, providing empirical support for the gradient-starvation effect (Prop. 2.2) where syntactic noise masks the topological signal. The transient spike at 5k reflects the volatility of the heuristic phase, where local approximations yield unstable alignment before collapsing. Subsequently, the monotonic rise from step 20k signifies Systematic Emergence. This steady consolidation validates Prop. 2.7, demonstrating that the interaction matrix successfully locks onto the low-rank semantic operator $M_{sem}$ after escaping the syntactic basin.

### 4.2. The Priority of Pointer Passing

To validate the prediction in Remark 2.10, we track the evolution of dependency resolution heads in Pythia-1.4B across steps 1k, 20k, and 143k, utilizing transitive identity chains (Natural Language) and value assignment chains

(code) as probes.

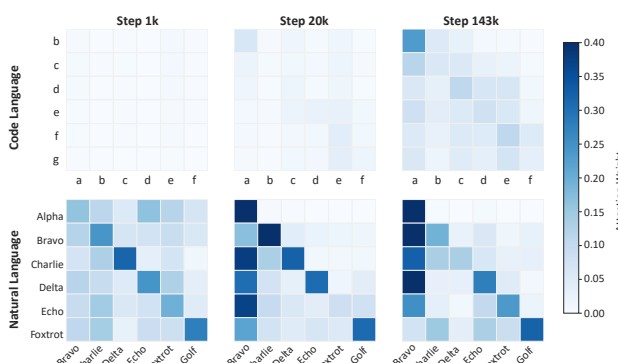

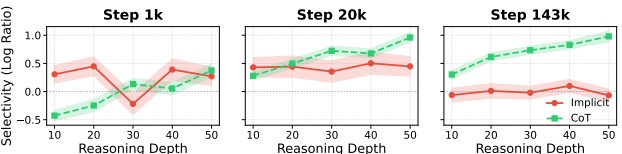

*Figure 7.* We report the log-ratio of gradient norms between the correct source and a distractor. Implicit (Red): Signal decays with depth, indicating long-range gradient starvation. CoT (Green): Intermediate supervision restores local gradients, stabilizing long-horizon learning.

*Figure 6.* We track the attention weights from query tokens (y-axis) to potential antecedents (x-axis) throughout the training of Pythia-1.4B. The probes utilize recursive variable assignments for code (top; e.g., `a = 42; b = a;`) and transitive identity chains for Natural Language (bottom; e.g., `The Alpha is the Bravo.`), visualizing the emergence of the pointer mechanism.

Figure 6 provides strong empirical support for our theory. In Natural Language, attention heads exhibit a stable diagonal structure throughout training, consistent with stepwise predecessor pointing. At Step 143k, the amplified first-column attention further indicates the emergence of multi-hop aggregation via iterative propagation.

In contrast, the code modality shows a delayed and shallower formation of this reference mechanism. We attribute this gap to stronger syntactic curvature in code, which raises the gradient barrier, and to reduced code exposure during pre-training, which prolongs signal accumulation.

### 4.3. Mechanistic Validation of CoT: Signal Restoration

To validate the parallel gradient pathway hypothesis (Prop. 2.11), we compare signal propagation in Implicit versus CoT modes within the training trajectory of Pythia-160m across steps 1k, 20k, and 143k. We utilize a gradient saliency probe to compute the *Selectivity Ratio* $\mathcal{R}$, defined as the proportion of gradient energy concentrated on the true causal ancestor $x_{src}$ relative to the total input sequence $\mathcal{S}$:

$$\mathcal{R} = \frac{\|\nabla_{x_{src}}\mathcal{L}\|_F}{\|\nabla_{\mathcal{S}}\mathcal{L}\|_F + \epsilon}. \quad (7)$$

This metric explicitly quantifies the model's sensitivity to the true causal ancestor, serving as a proxy for the alignment of the attention mechanism with the underlying computational topology.

The results in Figure 7 demonstrate a divergence in signal dynamics. Implicit reasoning (red) suffers from rapid signal decay as chain depth increases, confirming the downstream insensitivity predicted in Prop. 2.11. In contrast,

CoT (green) effectively short-circuits this bottleneck. By enforcing local stepwise prediction, CoT maintains high gradient selectivity regardless of total depth. This provides empirical evidence consistent with the hypothesis that externalizing reasoning traces may alleviate gradient starvation by restoring local learning signals.

## 5. Generalization to LLMs

To demonstrate the practical utility of our theoretical framework in realistic scenarios, we design a *Variable Cloze Completion* experiment where a specific variable usage in a code snippet is masked (e.g., `return a + b; return a + ;`) and the model must recover the correct identifier based on the context.

### 5.1. Experiment Settings

#### 5.1.1. MODELS.

We employ two open-weights models: Qwen2.5-Coder-7B (Hui et al., 2024; Yang et al., 2024) and Llama-3.1-8B (Dubey et al., 2024). These models represent standard production-scale architectures, allowing us to verify whether the mechanisms identified in our controlled probes scale to widely used foundation models.

#### 5.1.2. DATASETS.

We utilize Python150k and JavaScript150k datasets (Raychev et al., 2016). Crucially, these datasets provide native ASTs, which enables us to surgically target variable usage nodes for masking rather than relying on random token dropping. This ensures that the task strictly evaluates the model's ability to resolve semantic dependencies rather than relying on trivial local n-grams.

#### 5.1.3. BASELINES

To better position our method relative to existing structured learning approaches, we additionally compare against several representative baselines:

**Curriculum Learning** (Bengio et al., 2009), where samples are scheduled from easy to hard according to USE–DEF span distance;

**Linear-chain CRF** (Zheng et al., 2015), which imposes token-level sequential structural constraints over DEF/USE labels;

**Supervised Contrastive Learning (SupCon)** (Gunel et al., 2020), which aligns hidden representations of corresponding USE–DEF pairs;

**Attention Entropy Regularization** (Zhao et al., 2019), which encourages sharper attention distributions without directional semantic supervision.

### 5.2. Theoretic-Aligned Objective: Contrastive Attention as Phase Catalyst

Guided by our mechanistic framework, we construe the training objective not merely as likelihood maximization, but as a geometric intervention to optimize the alignment ratio $\rho(t)$ (Eq. 4). To mitigate the gradient starvation (Prop. 2.2) caused by syntactic noise, we introduce a topological auxiliary objective.

While the standard language modeling loss $\mathcal{L}_{LM}$ provides a diffuse signal, we explicitly incentivize the emergence of the semantic operator $M_{sem} = \mu_{use}\mu_{src}^T$. Let $\mathcal{H}_{tgt}$ be the set of semantic heads, and $\mathcal{N}$ be a sampled set of syntactic distractor indices. We formulate the contrastive objective as optimizing the margin between the source score $s_{src}$ and the average distractor score $\bar{s}_{neg} = \mathbb{E}_{n \in \mathcal{N}}[s_n]$:

$$\mathcal{L}_{total} = \mathcal{L}_{LM} + \lambda \sum_{h \in \mathcal{H}_{tgt}} \max\left(0, \eta - (s_{src}^{(h)} - \bar{s}_{neg}^{(h)})\right) \tag{8}$$

This formulation acts as a catalyst for the phase transition. Considering the gradient with respect to the interaction matrix $W_{QK}$ for an active constraint (i.e., where the margin is violated), the update direction acquires a specific rank-1 component:

$$\nabla_{W_{QK}}\mathcal{L}_{total} = \nabla_{W_{QK}}\mathcal{L}_{LM} \\ - \lambda \cdot \mathbb{I}_{active} \cdot \underbrace{\mu_{use}(\mu_{src} - \bar{\mu}_{neg})^T}_{\text{Steering Term}} \tag{9}$$

where $\mathbb{I}_{active}$ is the indicator function for margin violation, and $\bar{\mu}_{neg}$ is the centroid of the distractor embeddings. Unlike the opaque $\nabla\mathcal{L}_{LM}$, this steering term explicitly aligns the weight update with the semantic edge $\mu_{use}\mu_{src}^T$ while orthogonalizing it against syntactic directions $\mu_{use}\bar{\mu}_{neg}^T$. This mechanism effectively damps the curvature of the syntactic subspace relative to the semantic signal, thereby accelerating the traversal of the instability region (Phase II) and

facilitating a more robust convergence to the systematic reasoning regime.

### 5.3. Experiment Results

**Universality and Structural Sparsity.** To validate our framework's generality, we profiled attention dynamics across architectures (Llama-3.1-8B, Qwen2.5-Coder-7B) and languages (JavaScript, Python). We define the semantic binding strength $S_{l,h}$ for head $(l, h)$ as the expected attention mass from a usage token $u$ to its definition set $\mathcal{D}(u)$: $S_{l,h} = \mathbb{E}_u[\sum_{v \in \mathcal{D}(u)} A_{u,v}^{(l,h)}]$. As visualized in Figure 8, the results empirically confirm the Structural Sparsity predicted in Corollary 2.8: binding capabilities are concentrated in sparse, specialized heads rather than being diffusely distributed. This pattern remains invariant across models and syntaxes, as further evidenced by Qwen2.5-Coder-7B (see App. A.5).



*Figure 8.* **Existence and Universality of Semantic Binding.** Attention heatmaps ($S_{l,h}$) for Llama-3.1-8B on **(Left)** JavaScript and **(Right)** Python. The consistent emergence of these specific heads across syntactically diverse languages supports the existence of shared language-agnostic binding patterns across programming languages.

**Causal Verification via Head Ablation.** To confirm the functional role of these heads, we performed an ablation on the *Variable Cloze Completion* task by masking the Top-8 binding heads against a random-8 control. As shown in Table 1, while random masking caused a minor performance decline, ablating the binding heads led to a substantially larger drop. This significant gap suggests that these specific circuits play an important functional role in reference resolution.

**Contrastive Tuning and Results.** We fine-tuned the model using the osed alignment objective $\mathcal{L}_{total}$ with LoRA (Hu et al., 2022), applying the contrastive steering term exclusively to the Top-8 Binding Heads identified through profiling. This targeted intervention is designed to amplify the model's naturally emerging binding circuits, rather than imposing semantic structure on arbitrary components. As shown in Table 1, this geometrically aligned strategy yields a substantially larger performance gain than pure cross-entropy training: the improvement achieved by our method is more than twice the gain obtained from cross-entropy

*Table 1.* **Main Results on Variable Cloze Completion.** We report performance across two experimental settings. **Inference-time Ablation:** *Base* denotes the original model; *Top8* and *Random* refer to masking the identified binding heads versus a random selection to verify their functional importance. **Fine-tuning Methods:** *SFT* represents standard cross-entropy training; *CRF*, *Curriculum*, *SupCon*, and *Attn* denote the structured learning baselines described in Section 5.2; *Ours* utilizes the proposed topology-aligned contrastive loss.

| LANGUAGE | LLAMA-3.1-8B | | | | | | | | | QWEN2.5-CODER-7B | | | | | | | | |
|---|---|---|---|---|---|---|---|---|---|---|---|---|---|---|---|---|---|---|
| | BASE | TOP8 | RANDOM | SFT | CRF | CURR. | SUPCON | ATTN | OURS | BASE | TOP8 | RANDOM | SFT | CRF | CURR. | SUPCON | ATTN | OURS |
| PYTHON | 0.335 | 0.265 | 0.295 | 0.343 | 0.351 | 0.348 | 0.324 | 0.351 | **0.362** | 0.890 | 0.853 | 0.882 | 0.893 | 0.900 | 0.899 | 0.876 | 0.903 | **0.909** |
| JAVASCRIPT | 0.304 | 0.271 | 0.300 | 0.323 | 0.331 | 0.329 | 0.309 | 0.333 | **0.344** | 0.861 | 0.821 | 0.862 | 0.872 | 0.878 | 0.876 | 0.854 | 0.880 | **0.884** |

*Table 2.* **Generalization to Natural Language.** We evaluate whether the proposed topology-aligned objective extends beyond code by fine-tuning on natural-language coreference supervision and evaluating on WinoGrande.

| MODEL | BASE | SFT | OURS |
|---|---|---|---|
| QWEN2.5-7B | 0.679 | 0.845 | **0.874** |
| LLAMA-3.1-8B | 0.710 | 0.863 | **0.896** |

alone, suggesting that explicitly sharpening attention geometry can be more sample-efficient than implicit likelihood maximization in deep dependency learning tasks. All detailed experiment methods and settings can be seen in App. B.3.

**Cross-Language Generalization.** Finally, to test whether our method captures fundamental binding logic rather than surface-level syntax, we evaluate cross-lingual transfer. We apply the Python-tuned model to JavaScript and the JavaScript-tuned model to Python, without any additional fine-tuning. In both directions, our method consistently outperforms the baseline (see App. A.6). This cross-lingual generalization indicates that the Semantic Binding Circuits reinforced by our objective are abstract and language-agnostic, reflecting the shared topology of variable reference rather than language-specific syntax.

### 5.4. Generalization to Natural Language

To evaluate whether the proposed mechanism extends beyond code, we further study natural-language deep dependency learning through pronoun and coreference resolution. We map the notion of semantic dependency in code (i.e., USE–DEF relations) to antecedent–reference relations in natural language, where resolving a pronoun requires tracking latent relational structure across context.

Following this formulation, we construct training pairs using AllenNLP-based coreference extraction (Gardner et al., 2018; Lee et al., 2018). Given a reference mention (e.g., "his home"), we identify its antecedent span (e.g., "the house") and apply the same topology-aligned contrastive objective used in the code setting. To account for potential extraction noise in natural language, we additionally weight each pair by the confidence score returned by the coreference model.

We evaluate on the WinoGrande benchmark (Sakaguchi

et al., 2021), a widely used benchmark for pronoun resolution and deep contextual dependency tracking. Results are shown in Table 2. Across both Qwen2.5-7B and Llama3-8B, our method consistently improves over standard supervised fine-tuning, suggesting that the proposed optimization mechanism generalizes beyond code-specific structure.

## 6. Related Works

**Attention Mechanisms.** Existing research characterizes attention via mechanistic interpretability (Clark et al., 2019; Li et al., 2020; Jawahar et al., 2019; Olsson et al., 2022; Nanda et al., 2023; Conmy et al., 2023; Meng et al., 2022; Geva et al., 2021; Merullo et al., 2023; Rai et al., 2024), training dynamics (Tian et al., 2023; Ma et al., 2023; Dherin et al., 2025; Zhang et al., 2025; Teehan et al., 2022; Haviv et al., 2023; Geshkovski et al., 2023; Kobayashi et al., 2024), and theoretical connections to corpus statistics (Im et al., 2026; Dai et al., 2024; Prakash et al., 2024). Unlike these, we specifically address the unspecified conditions for realizing content-independent relational operators and reliable pointer-like binding.

## 7. Conclusion

In this work, we proposed a mechanistic framework for semantic emergence, characterizing it as a spectral competition where high-curvature syntax initially masks deep dependency learning via gradient starvation. Our experiments, spanning from few-layer toy models trained on synthetic data to production-scale LLMs, support the hypothesis that deep dependency learning can exhibit a sharp phase transition associated with the subspace alignment ratio. Crucially, we demonstrated that this process can be influenced through targeted interventions: methods such as CoT supervision and our topology-aligned contrastive objective help alleviate the syntactic barrier by rectifying gradient geometry. These findings motivate further exploration of pre-training strategies that move beyond simple likelihood maximization toward data curation and objectives that more explicitly encourage deep relational structure beyond surface statistical regularities.

## Acknowledgments

This work was supported by the State Key Lab. for Novel Software Technology (KFKT2024B06).

## Impact Statement

This paper presents work whose primary goal is to advance the field of Machine Learning by bridging the gap between mechanistic interpretability and model steering. By establishing a theoretical link between gradient dynamics and the emergence of reasoning circuits, we offer a framework for more transparent and controllable pre-training.

However, practical application involves trade-offs. First, due to circuit polysemanticity, attention heads in LLMs often encode multiple entangled features; aggressive fine-tuning of specific binding heads implies a risk of interfering with other latent capabilities. Second, our topology-aligned objective introduces computational overhead, as constraining internal activations significantly increases memory consumption (VRAM) during training compared to standard objectives. Finally, our experiments on LLMs are concentrated in the code domain. We utilize code because it offers explicit ground-truth for semantic dependencies (via ASTs), whereas real-world natural language lacks such unambiguous topological annotations. Future work is required to develop robust probing and steering methods that can handle the linguistic ambiguity inherent in human communication.

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

# A. More Experiment Results

All experiments in this section are performed with Qwen2.5-Coder-7B.

## A.1. Experimental Verification of Spectral Disparity

To validate the Spectral Disparity Assumption (2.1), we probe the local geometry of the loss landscape using *Iso-Energy Perturbations*. We hypothesize that the Hessian eigenspectrum is highly anisotropic, with syntactic directions corresponding to large eigenvalues (high curvature) and semantic directions corresponding to small eigenvalues (low curvature).

**Methodology.** We rely on the local quadratic approximation of the loss landscape. Around a local minimum, the change in loss under a perturbation $\delta$ is dominated by the Hessian $H$: $\Delta\mathcal{L} \approx \frac{1}{2}\delta^T H\delta$. To compare the spectral properties (curvature) of different subspaces, we apply perturbations $\delta = \alpha \cdot v$ along directions $v$ sampled strictly from the syntactic subspace ($\mathcal{U}_{syn}$) or the semantic subspace ($\mathcal{U}_{sem}$). Crucially, we enforce **normalization** ($\|v_{syn}\| = \|v_{sem}\| = 1$). Under this constraint, the loss change $\Delta\mathcal{L}(\alpha)$ becomes a direct proxy for the Rayleigh quotient $v^T H v$, which measures the curvature along direction $v$. Steeper parabolic growth implies larger Hessian eigenvalues.

**Results: Empirical Validation of Assumption 2.1.** The experimental results are visualized in Figure 9. The plot reveals a stark contrast in the local geometry of the two subspaces:

- **Syntactic Curvature (High $\lambda_{syn}$):** The loss landscape along syntactic directions (red curve) exhibits a sharp, steep valley. Quantitatively, a small perturbation of magnitude $|\alpha| = 0.2$ results in a substantial loss increase of $\Delta\mathcal{L} \approx 1.4$. This rapid growth indicates that the Hessian $H$ possesses large eigenvalues along these eigenvectors, confirming the high curvature of $\mathcal{U}_{syn}$.

- **Semantic Curvature (Low $\lambda_{sem}$):** In contrast, the landscape along semantic directions (blue curve) is remarkably flat, forming a "plateau." The loss increase is negligible ($\Delta\mathcal{L} < 0.2$) within the same perturbation range. This empirically confirms that the semantic subspace is associated with near-zero eigenvalues ($\lambda_{sem} \approx 0$).

**Conclusion.** The experiment yields a curvature ratio $\lambda_{syn}/\lambda_{sem} \gg 1$, providing direct physical evidence for the **Spectral Disparity** postulate in Assumption 2.1.

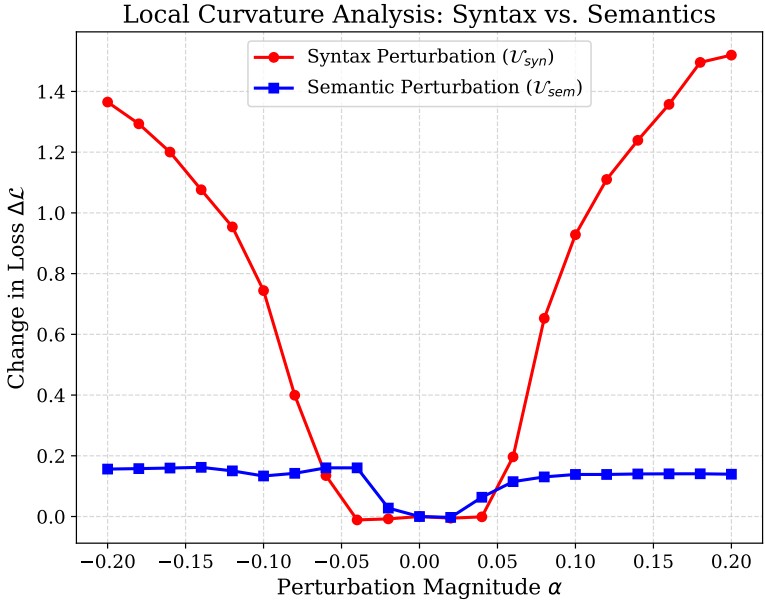

*Figure 9.* Loss landscape cross-section. The red curve (Syntax) shows significantly higher curvature (steeper valley) than the blue curve (Semantics), indicating $\lambda_{syn} \gg \lambda_{sem}$.

## A.2. Validation of Low-Rank Topological Operators

We experimentally test the core prediction of Prop. 2.7: that semantic binding circuits collapse into rank-1 topological operators ($M_{sem} \approx \mu_{use}\mu_{src}^T$), whereas syntactic processing remains high-rank.

**Methodology.** We analyze the singular value spectrum of the interaction matrices $W_{QK}^{(h)} \in \mathbb{R}^{d \times d}$ for all attention heads in Qwen2.5-Coder-7B. We utilize the **Stable Rank** (Effective Rank) to quantify the spectral concentration of the spectrum:

$$r_{eff}(W_{QK}) = \frac{\|\sigma\|_2^2}{\|\sigma\|_\infty^2} = \frac{\sum_k \sigma_k^2}{\sigma_{\max}^2}$$

A value $r_{eff} \to 1$ implies the matrix is dominated by a single principal component (Rank-1).

**Results.** The spectral analysis is visualized in Figure 10.

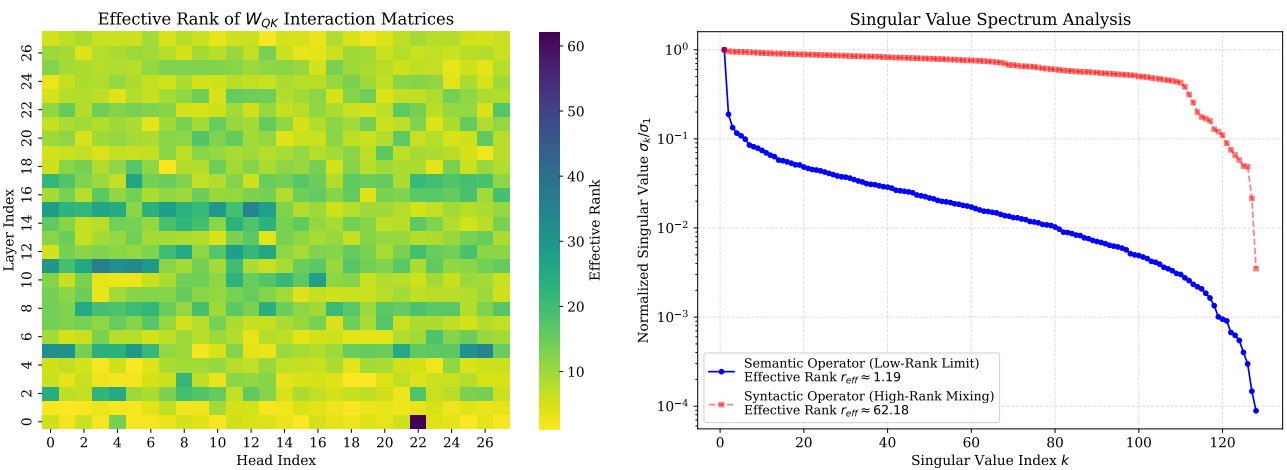

*Figure 10.* **Spectral Analysis of Interaction Matrices. Left:** Heatmap of effective rank across all layers and heads. The emergence of semantic operators is sparse; only specific heads (dark blue) collapse to the low-rank regime. **Right:** Singular value spectrum comparison. The Semantic Operator exhibits extreme spectral collapse with $r_{eff} \approx 1.19$, implying it functions as a precise Rank-1 pointer. The Syntactic Operator retains a high-dimensional mixing structure ($r_{eff} \approx 62.18$).

The empirical evidence strongly supports Prop. 2.7:

1. **Rank Collapse (Semantic):** As shown in the right panel (Blue Curve), the semantic head exhibits a single component dominated spectrum. The first singular value dominates the energy, resulting in an effective rank of $r_{eff} \approx 1.19$. This confirms that the learned matrix $W_{QK}$ has effectively converged to the rank-1 outer product $\mu_{use}\mu_{src}^T$.

2. **High-Dimensional Mixing (Syntactic):** In contrast, the syntactic head (Red Curve) shows a heavy-tailed spectrum with $r_{eff} \approx 62.18$, indicating diffuse information processing typical of distributed representations.

3. **Sparsity:** The heatmap (Left Panel) validates Corollary 3.9. Low-rank operators (dark blue pixels) are rare and localized, confirming that the binding mechanism is specialized to a sparse subset of heads.

## A.3. Validation of Systematic Gradient Emergence

Assumption 2.6 posits that the error signal $\gamma$ eventually decouples from context noise and aligns with a systematic semantic direction. To verify this denoising effect, we analyze the geometric coherence of gradient vectors across different layers of the model.

**Methodology.** We collect gradient vectors $g = \nabla_h \mathcal{L}$ with respect to the hidden states for specific semantic targets (e.g., variable assignment operations) across $N = 50$ diverse contexts. We employ two metrics to quantify gradient coherence:

1. **PCA Explained Variance (Top-1):** We compute the Principal Component Analysis of the gradient set $\{g_i\}_{i=1}^{N}$. A high explained variance ratio (EVR) for the first component indicates that the gradients are essentially 1-dimensional (pointing in a single semantic direction) rather than diffuse.

2. **Pairwise Cosine Similarity:** We measure the average cosine alignment $\frac{1}{N^2} \sum_{i \neq j} \cos(g_i, g_j)$. High similarity implies the learning signal is consistent across different contexts (context-independent).

**Results.** Figure 11 illustrates the evolution of gradient geometry across network depth.

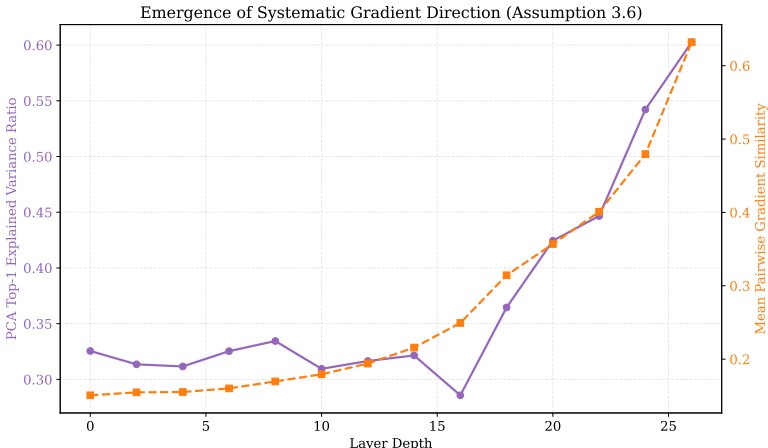

*Figure 11.* **Emergence of Systematic Gradient Direction.** We plot the geometric properties of the backpropagated error signal against layer depth. **purplePurple Line (PCA EVR):** The explained variance of the first principal component rises from $\approx 0.32$ in shallow layers to $> 0.60$ in deep layers. This indicates a *dimensionality collapse*: the error signal transitions from high-dimensional noise to a rank-1 systematic vector. **orangeOrange Line (Cosine Similarity):** The mean pairwise alignment increases monotonically, confirming that the gradient direction becomes robust to context noise as depth increases. This validates the noise-orthogonality postulate in Assumption 2.6.

The data confirms a phase transition in the error signal quality. In early layers ($L < 15$), gradients are noisy and context-dependent (low coherence). In deeper layers ($L > 20$), the gradients lock onto a single dominant direction. This Spectral Concentration ensures that the weight update $\Delta W \propto \gamma x^T$ accumulates constructively along the topological edge, enabling the emergence of the binding circuit.

### A.4. Visualizing the Variable Binding Subspace

Our theory predicts that semantic binding occurs within a specialized low-dimensional subspace where variable identities are disentangled from their values and contexts. To visualize this *Binding Subspace*, we extract hidden states corresponding to variable references and project them into 2D space.

**Methodology.** We generate $N = 1000$ random symbolic programs involving 26 variables ($a \ldots z$). We extract the activation vectors $h \in \mathbb{R}^{d_{model}}$ at the exact token positions where a variable is referenced. To isolate the binding-relevant geometry from the high-dimensional residual stream, we apply a two-step dimensionality reduction:

1. **Supervised Filtering:** We train a sparse linear probe (L1-regularized Logistic Regression) to predict the variable identity from $h$. We select the top $K = 30$ dimensions with the highest coefficients, effectively restricting the representation to the binding-relevant subspace.

2. **Manifold Projection:** We apply UMAP (Uniform Manifold Approximation and Projection) on these 30 dimensions to visualize the local topology.

**Results.** The projected manifold is shown in Figure 12.

The visualization confirms that the "semantic subspace" $\mathcal{U}_{sem}$ is not hypothetical but a concrete, extractable structure. The distinct islands for each variable demonstrate that the model has successfully factored the factor $P(variable|context)$ into a context-independent code, a prerequisite for the reliable pointer-passing mechanisms described in Remark 2.10.

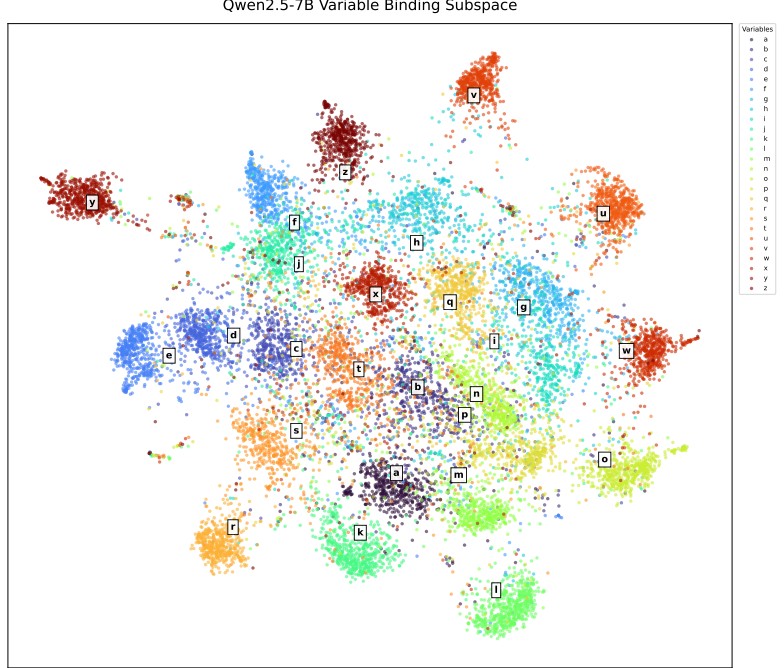

*Figure 12.* **The Geometry of Variable Binding.** We visualize the internal representation of 26 distinct variables ($a \ldots z$) at Layer 20 of Qwen2.5-7B. Each point represents a single occurrence of a variable in a unique random context. We observe clear, distinct **semantic clusters** for each variable (e.g., the red cluster 'v' at the top, the green cluster 'l' at the bottom). Despite the randomness of the surrounding code (contexts), the model maps all instances of the same variable to a tightly packed region in the binding subspace. This geometric disentanglement confirms that the model has learned abstract "Variable Types" as robust topological objects, independent of surface-level syntax.

### A.5. Structural Sparsity in Qwen

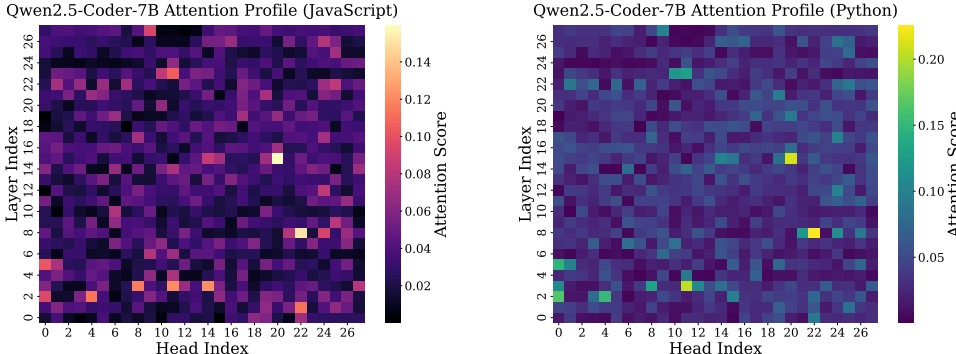

*Figure 13.* **Existence and Universality of Semantic Binding.** Attention heatmaps ($S_{l,h}$) for Qwen2.5-Coder-7B on (**Left**) JavaScript and (**Right**) Python.

### A.6. Cross-Language Generalization

The results are reported in Table 3.

## B. Experiment Settings

### B.1. Experiments in Sec. 3

We utilize a causal "NanoTransformer" architecture trained from scratch for all experiments. The model is a standard decoder-only Transformer with pre-normalization (LayerNorm applied before the self-attention and feed-forward blocks),

*Table 3.* Cross-language generalization performance. Models are trained on one programming language and evaluated on the other without additional fine-tuning. Our method consistently outperforms both Base and SFT across models and directions.

| TRAIN → TEST | METHOD | LLAMA | QWEN |
|---|---|---|---|
| PYTHON → JAVASCRIPT | BASE | 0.304 | 0.861 |
| | SFT | 0.300 | 0.861 |
| | OURS | **0.312** | **0.869** |
| JAVASCRIPT → PYTHON | BASE | 0.335 | 0.890 |
| | SFT | 0.337 | 0.887 |
| | OURS | **0.342** | **0.892** |

GeLU activation, and absolute positional embeddings. The vocabulary size is fixed at $V = 128$, partitioned into syntactic tokens (e.g., LET, =, ;, RETURN), variable identifiers ($v_0 \ldots v_{19}$), and numerical values.

All models were trained using the AdamW optimizer on a single NVIDIA 3080Ti GPU. To precisely control the dynamics, we employed a fixed seed (42) for the final reporting of results, particularly for the causal interventions in Experiment 3. The specific hyperparameters for each experimental setting are detailed in Table 4.

All runs in this experiment utilize the same initialization seed (42).

*Table 4.* Hyperparameter configurations for the three controlled experiments. Note that Experiment 2 uses a reduced model capacity to temporally extend the phase transition for better visualization.

| HYPERPARAMETER | EXP 1 (GRADIENT) | EXP 2 (CURRICULUM) | EXP 3 (CAUSAL) |
|---|---|---|---|
| LAYERS ($L$) | 4 | 3 | 4 |
| MODEL DIM ($d_{model}$) | 128 | 64 | 128 |
| ATTN HEADS ($H$) | 4 | 4 | 4 |
| SEQ LENGTH ($T$) | 64 | 48 | 48 |
| LEARNING RATE ($\eta$) | $1.0 \times 10^{-3}$ | $8.0 \times 10^{-4}$ | $1.0 \times 10^{-3}$ |
| BATCH SIZE | 128 | 128 | 128 |
| MAX STEPS | 5000 | 3500 | 1500 |

**Experiment 1: Visualizing Gradient Starvation.** The objective of this experiment is to monitor the L2 norms of gradients at the embedding layer during the initial learning phase. The dataset generator produces *Pointer Binding* sequences involving variable assignment (e.g., LET v1 = 5;) and retrieval (RETURN v1). We define two token groups for monitoring: $\mathcal{T}_{syn}$ comprising syntax tokens (indices $0 \ldots 9$) and $\mathcal{T}_{sem}$ comprising variable identifiers (indices $10 \ldots 29$). We record the average gradient norm $\|\nabla_E \mathcal{L}\|_2$ for these subgroups at every step to visualize the delay in semantic learning relative to syntactic structure.

**Experiment 2: Escaping Softmax Saturation (Curriculum).** To investigate the how early attention concentration on syntactic tokens suppresses semantic learning and subsequent release, we employ a curriculum learning setup. The model is initially trained on a variable assignment (copy) task. At step $t = 1500$, the data generator switches to an arithmetic addition task (RETURN v1 + v2). To capture the phase transition mechanics, we use a smaller model capacity ($d_{model} = 64$) and a reduced learning rate ($8 \times 10^{-4}$). We monitor the attention weights $A_{QK}$ of the last layer, specifically tracking the attention mass allocated to operand tokens versus syntactic anchors. High-frequency sampling is applied around the task switch (steps 1475–1600) to resolve the crossover dynamics.

**Experiment 3: Causal Validation via Intervention.** This experiment validates the causal role of the alignment ratio $\rho(t)$ using a *Double Dissociation* protocol. We compare a standard control run against two intervention conditions using a Moderate Noise Generator. (1) **Syntax-Suppressed:** We artificially mask the gradients of syntactic tokens at the embedding layer ($\nabla_{syn} \leftarrow 0$) during backpropagation. This intervention removes the "noise" from the syntactic subspace, theoretically increasing $\rho(t)$ and accelerating learning. (2) **Variable-Disturbed:** We inject Gaussian noise into the gradients of variable tokens ($\nabla_{var} \leftarrow \nabla_{var} + \mathcal{N}(0,1)$). This perturbation disrupts the semantic alignment direction $M_{sem}$, decreasing $\rho(t)$ and delaying the emergence of reasoning.

*Table 5.* Model specifications and checkpoint granularity. Exp 1 uses dense sampling to capture the phase transition curve, while Exp 2 and 3 focus on snapshot comparisons.

| PARAMETER | EXP 1: ALIGNMENT | EXP 2: POINTERS | EXP 3: COT |
|---|---|---|---|
| MODEL | PYTHIA-160M | PYTHIA-1.4B | PYTHIA-160M |
| HIDDEN DIM | 768 | 2048 | 768 |
| SAMPLING | DENSE (7 STEPS) | SPARSE (3 STEPS) | SPARSE (3 STEPS) |
| CHECKPOINTS | 1K, 5K, . . . , 143K | 1K, 20K, 143K | 1K, 20K, 143K |
| PRECISION | FP16 | FP16 | FP16 |

## B.2. Experiments in Sec. 4

We detail the experimental setup for the three mechanistic analyses performed on the Pythia model family. All experiments were conducted using the HuggingFace `transformers` library with `GPTNeoXForCausalLM` architectures. The analysis spans a training trajectory from initialization to convergence, utilizing Pythia-160m for gradient-sensitive probes and Pythia-1.4b for structural attention analysis.

**Model Configuration and Sampling.**    To capture the distinct phases of emergence, we employ different checkpoint granularities. For the alignment dynamics (Exp 1), we use a dense sampling strategy to capture the phase transition. For the structural and CoT analyses (Exp 2 & 3), we utilize three pivotal checkpoints ($t \in \{1\text{k}, 20\text{k}, 143\text{k}\}$) representing the heuristic phase, the transition point, and convergence. Table 5 summarizes these configurations.

**Exp 1: Probing the Alignment Ratio $\rho(t)$.**    To validate the subspace competition hypothesis, we quantify the geometric alignment between the learned attention mechanism and the ideal semantic operation. We define the alignment ratio $\rho(t)$ as the normalized projection of the model's query-key interaction matrix $W_{QK}^{(t)}$ onto the ground-truth semantic operator $M_{sem}$:

$$\rho(t) = \frac{\langle W_{QK}^{(t)}, M_{sem}\rangle_F}{\|W_{QK}^{(t)}\|_F \|M_{sem}\|_F} \tag{10}$$

The matrix $M_{sem}$ is constructed as a low-rank operator encoding the ideal dependency structure of the synthetic grammar. We compute this metric across dense checkpoints to visualize the non-monotonic transition from gradient starvation (where $\rho \approx 0$) to semantic locking.

**Exp 2: Retrospective Head Matching.**    We track the evolution of pointer mechanisms by generating synthetic dependency tasks (Table 6). To isolate the relevant circuit, we employ a "Time Travel" protocol. We first identify the "Process Head" at the final checkpoint (Step 143k) that maximizes attention mass on the correct antecedent tokens ($L^*, H^*$). We then load earlier checkpoints (1k, 20k) and extract the attention patterns from this specific head indices, ensuring we are tracing the evolution of a fixed structural component rather than dynamically selecting the strongest head at each step.

**Exp 3: Gradient Saliency and CoT Restoration.**    To verify the signal restoration hypothesis, we measure the gradient flow through the network using a Gradient Saliency Probe. For each checkpoint $t \in \{1\text{k}, 20\text{k}, 143\text{k}\}$, we compute the gradient of the correct target logit with respect to the input embeddings $E$. The selectivity score is defined as the log-ratio of the gradient norm at the causal source $x_{src}$ versus a distractor $x_{dist}$. We generate $N = 60$ unique random graph instances for each depth $d \in \{10, \ldots, 50\}$. In the Implicit setting, the model must attend directly to the distant root value (distractor: interleaved chain value); in the CoT setting, the model is prompted to identify the immediate parent (distractor: root value), testing its ability to utilize local gradient pathways.

## B.3. Our Loss

To enforce the mechanistic alignment between the model's latent topology and the code's semantic structure, we propose a composite objective function. This objective combines a syntax-aware generation loss with a geometric constraint on the attention mechanism.

**Variable-Weighted Causal Modeling.**    Standard language modeling treats all tokens equally, often allowing the model to minimize loss via trivial syntactic patterns (e.g., punctuation) rather than resolving long-range dependencies. To counteract

*Table 6.* Procedural generation templates for mechanistic probes. Random variable names (e.g., x9k) are used in Exp 3 to prevent token frequency artifacts.

| EXP | TASK STRUCTURE | TEMPLATE / LOGIC |
|---|---|---|
| EXP 2 | NATURAL LANGUAGE (TRANSITIVE IDENTITY) | THE ALPHA IS THE BRAVO. QUERY: # THE ALPHA IS THE → TARGET: BRAVO |
| | CODE (VALUE ASSIGNMENT) | A = 42; B = A; C = B; QUERY: # VALUE OF B IS → TARGET: A |
| EXP 3 | HARD MODE SHUFFLE | INTERLEAVED CHAINS $(A, B)$ WITH RANDOM VARS. |
| | IMPLICIT PROBE | CONTEXT: ... x9K=Q2Z. Q: WHAT IS x9K? TARGET: ROOT VALUE (LONG-RANGE JUMP) |
| | COT PROBE | CONTEXT: ... x9K=Q2Z. Q: x9K IS COPY OF? TARGET: Q2Z (LOCAL HOP) |

this, we introduce a re-weighted cross-entropy loss $\mathcal{L}_{\text{LM}}$. We define a weight vector $\mathbf{w} \in \mathbb{R}^T$ for a sequence of length $T$, where $w_t = \alpha$ if $x_t$ belongs to a variable identifier (definition or usage), and $w_t = 1$ otherwise. Based on our preliminary profiling, we set $\alpha = 5.0$ to force the gradient descent to prioritize variable binding accuracy:

$$\mathcal{L}_{\text{LM}} = -\frac{1}{\sum w_t} \sum_{t=1}^{T} w_t \log P(x_t \mid x_{<t}; \theta) \tag{11}$$

**Contrastive Attention Alignment.** To explicitly construct the retrieval circuit, we impose a hinge loss on a subset of "target heads" $\mathcal{H}_{target}$. These heads are selected based on their pre-trained alignment scores (top $K = 8$ heads). For a given variable usage token $q_{use}$, we require the attention mechanism to prioritize the corresponding definition token $k_{def}$ over random negative tokens $k_{neg}$. The alignment loss $\mathcal{L}_{\text{Align}}$ is defined as:

$$\mathcal{L}_{\text{Align}} = \frac{1}{|\mathcal{H}|} \sum_{h \in \mathcal{H}} \mathbb{E}_q \left[ \max \left( 0, \gamma - (A_h(q_{use}, k_{def}) - A_h(q_{use}, k_{neg})) \right) \right] \tag{12}$$

where $A_h(\cdot)$ represents the attention weight, and we set the margin $\gamma = 0.2$. To prevent training instability, the weight of this term $\lambda(t)$ follows a linear warmup schedule, peaking at $\lambda_{max} = 0.5$.

**Dynamic Role Inference.** Since the raw C# training corpora lack explicit semantic labeling for "Definition" versus "Usage," we implement a heuristic inference logic during data loading. For every unique identifier, we sort its occurrences by position; the first occurrence is labeled as DEF, and all subsequent occurrences are labeled as USE. This allows us to construct the $(q_{use}, k_{def})$ pairs required for $\mathcal{L}_{\text{Align}}$ without requiring an external static analysis compiler.

**Implementation and Training Details.** We fine-tune Qwen2.5-Coder-7B using Low-Rank Adaptation (LoRA) to minimize memory overhead. We target all linear projection layers ($W_q, W_k, W_v, W_o$ and FFN gates) with rank $r = 16$ and $\alpha = 32$. The model is trained with a batch size of 1 and gradient accumulation steps of 32 (effective batch size 32). We use the AdamW optimizer with a learning rate of $2 \times 10^{-5}$ and a cosine decay schedule with 10% warmup. The maximum sequence length is set to 2048 tokens to accommodate long-context variable dependencies.

# C. Detailed Derivations and Theoretical Proofs

In this appendix, we provide the formal derivations and strict assumptions for the mechanistic claims made in Section 2.

## C.1. Formal Analysis of Gradient Starvation (Supplementary to Sec. 2.2)

We rigorously justify Prop. 2.2 by defining the residual distribution and applying spectral bounds.

### C.1.1. ASSUMPTION ON RESIDUAL ISOTROPY

To formally compare gradient projections, we assume the initial residual $e = \theta - \theta^*$ is isotropic up to the second moment.

**Assumption C.1** (Second-Order Isotropy)**.** We assume the parameter residual $e$ follows a distribution $\mathcal{D}_e$ with zero mean and scaled identity covariance:

$$\mathbb{E}[e] = 0, \quad \mathbb{E}[ee^T] = \sigma_e^2 I \tag{13}$$

Consequently, for any projection operator $P$ onto a subspace of dimension $m$, $\mathbb{E}[\|Pe\|^2] = \sigma_e^2 m$.

### C.1.2. PROOF OF STARVATION RATIO

Consider the quadratic approximation $\mathcal{L}(\theta) \approx \mathcal{L}(\theta^*) + \frac{1}{2}e^T H e$. The gradient is $g = He$. We analyze the expected squared gradient norm in the syntactic subspace $\mathcal{U}_{syn}$ spanned by eigenvectors with eigenvalues $\{\lambda_k\}_{k \in \mathcal{I}_{syn}}$.

*Proof.* Using the eigendecomposition $H = \sum_k \lambda_k v_k v_k^T$:

$$\mathbb{E}[\|g_{syn}\|^2] = \sum_{k \in \mathcal{I}_{syn}} \lambda_k^2 \cdot \mathbb{E}[(v_k^T e)^2] = \sigma_e^2 \sum_{k \in \mathcal{I}_{syn}} \lambda_k^2 \tag{14}$$

Applying spectral bounds with $\lambda_{syn}^{\min} = \min_{k \in \mathcal{I}_{syn}} \lambda_k$:

$$\mathbb{E}[\|g_{syn}\|^2] \geq m_{syn}(\lambda_{syn}^{\min})^2 \sigma_e^2 \tag{15}$$

Assuming spectral disparity $\lambda_{syn}^{\min} \gg \lambda_{sem}^{\max}$, the ratio of expected gradient norms satisfies:

$$\frac{\mathbb{E}[\|g_{syn}\|^2]}{\mathbb{E}[\|g_{sem}\|^2]} \geq \frac{m_{syn}}{m_{sem}} \left(\frac{\lambda_{syn}^{\min}}{\lambda_{sem}^{\max}}\right)^2 \gg 1 \tag{16}$$

This confirms that the gradient energy is dominated by the syntactic subspace. $\square$

## C.2. Derivation of the Vanishing Gradient Barrier (Supplementary to Sec. 2.3)

We derive the exponential decay bound using the correct chain rule.

### C.2.1. DERIVATIVE VIA CHAIN RULE

The gradient w.r.t. a score $s_{sem}$ is given by the Jacobian of the softmax $A(s)$:

$$\frac{\partial \mathcal{L}}{\partial s_{sem}} = \sum_k \frac{\partial \mathcal{L}}{\partial A_k} \frac{\partial A_k}{\partial s_{sem}} = A_{sem}\left(\frac{\partial \mathcal{L}}{\partial A_{sem}} - \sum_k A_k \frac{\partial \mathcal{L}}{\partial A_k}\right) \tag{17}$$

### C.2.2. EXPONENTIAL DECAY UNDER BOUNDEDNESS

**Assumption C.2** (Bounded Downstream Gradient)**.** We assume the downstream gradient is bounded: $\left|\frac{\partial \mathcal{L}}{\partial A_k}\right| \leq C$.

*Proof.* Let $D_k = \frac{\partial \mathcal{L}}{\partial A_k}$. The term in parenthesis is bounded by $2C$. In the Top-1 syntactic domination regime ($s_{syn} \gg s_{sem}$), we have $A_{sem} \approx e^{-(s_{syn} - s_{sem})}$. Thus:

$$\left|\frac{\partial \mathcal{L}}{\partial s_{sem}}\right| \leq 2C \cdot A_{sem} = O\left(e^{-(s_{syn} - s_{sem})}\right) \tag{18}$$

This confirms the exponential vanishing of the semantic learning signal. $\square$

## C.3. Hebbian Reconstruction Analysis (Supplementary to Sec. 2.4)

We rigorously justify the rank-1 update direction by analyzing the sensitivity of the error signal to noise.

### C.3.1. TEACHER GRAPH AND ERROR SIGNAL SIGNS

We assume a generative model where $y_j = f(x_{i^*(j)})$ and $i^*(j)$ is the unique true dependency. The parameter update is proportional to $-\nabla_W \mathcal{L} \approx -\sum_{j,i} \gamma_{ji} x_j x_i^T$. For the correct edge, increasing attention decreases loss, so $\mathbb{E}[\gamma_{ji^*}] = -c$ with $c > 0$. For irrelevant edges, $\mathbb{E}[\gamma_{ji}] \approx 0$.

### C.3.2. NOISE SENSITIVITY AND RANK-1 DOMINANCE

We decompose embeddings as $x = \mu + \delta$, where $\delta$ is zero-mean noise with covariance $\Sigma_\delta$. We assume the error signal $\gamma(x)$ is locally smooth w.r.t. noise.

**Assumption C.3** (Weak Noise Sensitivity). We approximate $\gamma$ via a first-order Taylor expansion around the signal $\mu$:

$$\gamma(\mu + \delta) \approx \gamma(\mu) + \nabla\gamma(\mu)^T \delta \tag{19}$$

This implies that the correlation between error and noise is of second order: $\mathbb{E}[\gamma\delta] \approx O(\mathbb{E}[\|\delta\|^2])$.

Using this expansion, the expected update direction becomes:

$$\mathbb{E}[-\nabla_W \mathcal{L}] = \mathbb{E}\left[ \sum_j -(\gamma(\mu_j) + \epsilon)(\mu_j + \delta_j)(\mu_{i^*} + \delta_{i^*})^T \right]$$
$$= \sum_j c \cdot \mu_j \mu_{i^*}^T + O(\mathbb{E}[\|\delta\|^2]) \tag{20}$$

The cross-terms (e.g., $\gamma\mu\delta^T$) vanish in expectation due to the independence of $\delta$ across positions or its zero mean. Thus, the update is dominated by the rank-1 topological operator $\mu_{use}\mu_{src}^T$, with noise terms acting as a second-order perturbation.

### C.4. Derivation of the Alignment Threshold (Supplementary to Sec. 2.5)

We derive the stability condition for $\rho(t)$. Let the gradient be $\nabla = S + N$, where $S$ is the systematic semantic direction and $N$ is zero-mean noise. We explicitly assume the noise is uncorrelated with the signal in expectation:

$$\mathbb{E}[\langle S, N \rangle] = 0 \tag{21}$$

The expected alignment ratio (cosine similarity squared) is:

$$\mathbb{E}[\rho] \approx \frac{\|S\|^2}{\mathbb{E}[\|\nabla\|^2]} = \frac{\|S\|^2}{\|S\|^2 + \mathbb{E}[\|N\|^2] + 2\mathbb{E}[\langle S, N \rangle]} = \frac{1}{1 + \text{SNR}^{-1}} \tag{22}$$

where $\text{SNR} = \|S\|^2 / \mathbb{E}[\|N\|^2]$. The threshold $\tau_{stable}$ corresponds to the critical SNR where the systematic signal $S$ dominates the variance of $N$.

### C.5. Formal Argument for Pointer Priority (Supplementary to Remark 2.10)

We formalize the precedence of Pointer $(i \to j)$ over Value $(i \to k)$ based on sample complexity and signal accumulation.

### C.5.1. MECHANISM VS. FACT

- Pointer (Mechanism): Relies on Type-level consistency ($N_{type}$ samples).

- Value (Fact): Relies on Instance-level consistency ($N_{inst}$ samples).

Clearly $N_{type} \gg N_{inst}$.

### C.5.2. SNR SCALING LAW

We model the learning process as signal accumulation under additive Gaussian noise. Let $g$ be the per-sample gradient. The cumulative update over $N$ samples is $\Delta W \sim \sum_{n=1}^{N} (g_{signal} + \xi_{noise})$.

- Signal magnitude: $\|\sum g_{signal}\| \propto N$

- Noise magnitude (Random Walk): $\|\sum \xi_{noise}\| \propto \sqrt{N}$

The effective Signal-to-Noise Ratio for the learned circuit scales as:

$$\text{SNR}(N) = \frac{\text{Signal}}{\text{Noise}} \propto \frac{N}{\sqrt{N}} = \sqrt{N} \tag{23}$$

Comparing the Pointer and Value circuits:

$$\frac{\text{SNR}_{ptr}}{\text{SNR}_{val}} \approx \sqrt{\frac{N_{type}}{N_{inst}}} \gg 1 \tag{24}$$

Since $\text{SNR}_{ptr}$ grows significantly faster, $\rho_{ptr}(t)$ crosses the stability threshold $\tau_{stable}$ earlier than $\rho_{val}(t)$, ensuring the pointer mechanism emerges first.

### C.6. Accelerating Semantic Emergence via CoT

While CoT is widely used for complex reasoning, its mechanistic basis remains under-explored. We show that CoT mitigates the Softmax Saturation barrier (Thm. 2.4) by making intermediate states explicit. At inference time, explicit traces expose latent states as context tokens, improving semantic retrieval. During training, supervising these traces adds a local loss term that provides a parallel gradient pathway, alleviating credit assignment bottlenecks.

**Problem Formulation.**  Consider a compositional task $x \rightarrow y$ mediated by an intermediate sequence $z$ (e.g., $z$ is the resolved binding/value-fetch result, and $y$ is the final output).

- **Implicit Reasoning:** The model outputs $y$ directly, i.e., $P(y \mid x)$, while the intermediate $z$ remains a latent computation.

- **CoT Prompting (Inference-time):** The model generates the joint sequence $(z, y)$, i.e., $P(z, y \mid x) = P(z \mid x) \, P(y \mid x, z)$.

**Inference-time CoT: Externalization as State Injection.**  Implicit reasoning must resolve multi-hop dependencies (e.g., $x \rightarrow z \rightarrow y$) within a single forward pass. Under softmax saturation (Thm. 2.4), attention can be dominated by syntactic tokens, making semantic predecessor retrieval unreliable and degrading $P(y \mid x)$. CoT prompting externalizes the intermediate state $z$ as an explicit token, converting the hard conditional $P(y \mid x)$ into two easier conditionals $P(z \mid x)$ and $P(y \mid x, z)$, which stabilizes downstream semantic attention.

**Training-time CoT Supervision: Objective-level Injection.**  We now analyze the regime where intermediate traces $z$ are supervised during training, as in CoT supervised fine-tuning or CoT distillation. In this case, the training objective factorizes as:

$$\begin{aligned}
\mathcal{L}_{\text{CoT-sup}} &= -\log P(z, y \mid x) \\
&= -\log P(z \mid x) - \log P(y \mid x, z) \\
&= \mathcal{L}_z + \mathcal{L}_{y|z}.
\end{aligned} \tag{25}$$

Unlike inference-time prompting, this objective introduces an explicit loss term $\mathcal{L}_z$ that directly supervises the intermediate state $z$, creating a parallel and more local gradient signal for binding-related parameters.

**Proposition C.4** (Gradient Attenuation in Implicit Reasoning). *In the implicit setting, the loss $\mathcal{L}_{imp} = -\log P(y \mid x)$ depends on the binding-circuit parameters $\theta_{sem}$ only through the downstream computation $y = g(z)$. Let $s_{sem}$ denote the binding-relevant score (e.g., an attention logit). Using Jacobian notation (denominator layout), the chain rule gives:*

$$\begin{aligned}
b &:= J_y(z)^\top \, \nabla_y \mathcal{L}_{imp}, \\
\nabla_{\theta_{bind}} \mathcal{L}_{imp} &= J_{s_{bind}}(\theta_{bind})^\top \, J_z(s_{bind})^\top \, b.
\end{aligned} \tag{26}$$

*where $J_a(b) := \frac{\partial a}{\partial b}$. This path is vulnerable to two failure modes:*

1. **Downstream Insensitivity:** *If the Jacobian $J_y(z) = \frac{\partial y}{\partial z}$ has small spectral norm, the signal can vanish before reaching $\theta_{sem}$.*

2. **Upstream Gating:** *As shown in Lemma 2.3, the local derivative $J_z(s_{bind}) = \frac{\partial z}{\partial s_{bind}}$ is multiplicatively suppressed under softmax saturation. In the syntactic phase, this term can effectively gate the entire gradient chain.*

**Proposition C.5** (Parallel Gradient Pathway under CoT Supervision). *Under training-time CoT supervision, the total gradient becomes an additive composition:*

$$\nabla^{total}_{\theta_{bind}} = \nabla_{\theta_{bind}}\mathcal{L}_{y|z} + \underbrace{\nabla_{\theta_{bind}}\mathcal{L}_z}_{\text{Injected Gradient}} . \tag{27}$$

*The injected term $\nabla_{\theta_{bind}}\mathcal{L}_z$ depends only on predicting the intermediate state $z$. It provides a parallel learning signal that bypasses the downstream sensitivity term $J_y(z)$ and remains active even when the implicit pathway is weak or gated by saturation. When $\|\nabla_{\theta_{bind}}\mathcal{L}_{y|z}\|$ is near-zero due to starvation/saturation, this additional pathway effectively short-circuits the optimization bottleneck.*

**Corollary C.6** (Semantic Alignment Boost under CoT Supervision). *Recall the semantic operator $M_{sem}$ defined in Section 2.5. Let $\Delta^{imp}$ denote the implicit gradient component from $\mathcal{L}_{y|z}$, and let $\Delta^{inj}$ denote the injected gradient from $\mathcal{L}_z$. Assuming the supervised traces are* valid*, i.e., $z$ encodes dependency edges that causally support the correct output $y$, we have*

$$\langle\Delta^{inj}, M_{sem}\rangle_F \gg 0. \tag{28}$$

*Therefore the total semantic projection increases additively:*

$$\langle\Delta^{total}, M_{sem}\rangle_F = \langle\Delta^{imp}, M_{sem}\rangle_F + \langle\Delta^{inj}, M_{sem}\rangle_F, \tag{29}$$

*This increases the semantic projection of the update direction and thus tends to raise the alignment ratio $\rho(t)$ in Equation (4). As a result, CoT supervision accelerates the crossing of the stability threshold $\tau_{stable}$.*

*Remark* C.7 (Why CoT Helps Reasoning More Than Pattern Matching). This framework explains why CoT supervision yields large gains on deep compositional tasks but only marginal improvements on shallow pattern matching. Define the *Relative Semantic Gain*:

$$\text{Gain} := \frac{\left\|P_{sem}(\nabla^{imp} + \nabla^{inj})\right\|_F}{\left\|P_{sem}(\nabla^{imp})\right\|_F + \epsilon}. \tag{30}$$

By subadditivity of the norm,

$$\text{Gain} \leq 1 + \frac{\left\|P_{sem}(\nabla^{inj})\right\|_F}{\left\|P_{sem}(\nabla^{imp})\right\|_F + \epsilon}. \tag{31}$$

When $P_{sem}(\nabla^{inj})$ is positively aligned with $P_{sem}(\nabla^{imp})$, this bound is approximately tight, yielding

$$\text{Gain} \approx 1 + \frac{\left\|P_{sem}(\nabla^{inj})\right\|_F}{\left\|P_{sem}(\nabla^{imp})\right\|_F + \epsilon}. \tag{32}$$

- **Deep Compositional Tasks:** The implicit semantic signal is weak due to starvation/saturation, so $\|P_{sem}(\nabla^{imp})\|_F$ is small. Injected gradients dominate, yielding Gain $\gg 1$ and accelerating the phase transition.

- **Shallow Pattern Matching:** The task does not rely strongly on the semantic subspace, so $\left\|P_{sem}(\nabla^{inj})\right\|_F$ is also small, yielding Gain $\approx 1$.

