# OpenReview forum: "Syntax vs. Semantics: How Transformers Learn Deep Dependencies"
_ICML.cc/2026/Conference — ICML 2026 regular_

### Official Review · Reviewer_QRWU · 2026-03-04

**Soundness:** 3
**Presentation:** 3
**Significance:** 3
**Originality:** 3
**Overall Recommendation:** 4
**Confidence:** 4

**Summary:**

This paper aims to provide a mechanistic perspective on how Transformer-based language models acquire semantic dependencies beyond surface-level syntactic patterns. The authors argue that training involves two types of signals: surface statistics and deep semantics. In early stages, surface features dominate due to stronger gradient signals, leading the model to prioritize simpler patterns while semantic learning remains comparatively weak. This phenomenon is described as gradient starvation. The paper further suggests that, after a certain point, some attention heads specialize rather abruptly, enabling improved modeling of long-range dependencies. While this transition is interesting, it would be helpful to clarify how consistently it appears across different model sizes and training settings.

To track this process, the authors introduce an alignment ratio as a proxy for semantic reliance. Results on both synthetic and larger-scale tasks suggest that semantic improvements coincide with a small number of specialized attention heads, though the robustness of this metric may require further discussion. The paper also links these findings to the effectiveness of CoT prompting, arguing that it may help redirect gradients toward semantic features; however, the connection between prompting and training dynamics could be better justified. Finally, a contrastive fine-tuning approach based on abstract syntax trees is proposed and shows improved performance on tasks such as variable binding. Overall, the paper offers an interesting perspective by framing semantic generalization as a matter of training dynamics, but several claims would benefit from further empirical support and clarification.

**Compliance With Llm Reviewing Policy:**

Affirmed.

**Final Justification:**

The additional experiments provided by the authors in their rebuttal—such as comparisons with baselines like CRF, curriculum learning, and supervised contrastive learning—effectively addressed my main concerns. My evaluation of the paper has therefore improved.

**Key Questions For Authors:**

1. On natural language datasets, does the alignment ratio exhibit a similar "phase transition"?

2. The experiments (Exp1–Exp3 in Table 4) use models with different parameters and hidden dimensions. I’m not entirely sure what motivated this design choice. Could the authors clarify how these variations affect the observed trends, and whether the main conclusions remain consistent across model sizes?

3. If the paper gets accepted, would you be open to releasing the code and operation script?

**Limitations:**

yes

**Strengths And Weaknesses:**

Strengths:

1. The authors introduce gradient starvation as a plausible explanation for the observed phenomenon where surface features dominate early training while semantic learning lags behind.

2.By defining the alignment ratio $\rho (t)$, the authors characterize phenomena such as grokking as a quantifiable and partially predictable phase transition.

3.The discussion of chain-of-thought prompting from a training dynamics perspective offers a possible causal interpretation.

4.The overall structure of the paper is clear and relatively easy to follow.

Weaknesses:

1.The distinction between “surface statistics” and “deep semantics” is fairly clear in structured domains like code (e.g., via ASTs), but much less so in natural language. It is therefore not entirely clear how well the proposed mechanism carries over to more realistic settings.

2.The analysis focuses mainly on attention dynamics, with relatively little discussion of other components (such as MLP layers or residual connections), which could also contribute to semantic learning.

3 There is limited comparison with other approaches that aim to promote structured learning or address similar training challenges (e.g., curriculum learning or structured objectives), which makes it harder to evaluate the relative benefits of the proposed method.

4. Some of the analysis methods (e.g., layer-wise gradient tracking, stable rank computation, and alignment ratio monitoring) appear relatively costly.
I am not sure how practical these tools would be at the scale of modern large models (e.g., billion- or trillion-parameter settings), especially in standard training or debugging workflows.

---

> ### Author Rebuttal · Authors · 2026-03-29
>
> We sincerely thank the reviewer QRWU for the thoughtful evaluation, constructive feedback, and recognition of the clarity and coherence of our work. Additionally, the reviewer raised several important concerns and questions, which we address in detail below.
>
> > **Q1:** Generalization to natural language datasets.
>
> **R1: 1. Natural Language Phase Transition:** First, Section 4.1 of our paper is already conducted on natural language (rather than code), where we observe the same phase transition behavior of the alignment ratio. We will clarify this more explicitly in the revised version.
>
> **2. Extended NL Causal Validation:** To further validate applicability to natural language, we extend Section 3.3 (Causal Validation of alignment ratio Thresholds) with additional NL experiments. In the updated figure, the orange curve represents complete removal of syntactic elements, addressing the concern raised by reviewer QSVG (Q1). The results show that the behavior on natural language is fully consistent with that observed in code. Results: https://anonymous.4open.science/r/Rebuttal-for-ICML2026/QSVG/R2.pdf
>
> **3. NL Fine-tuning via Coreference Resolution:** For LLM fine-tuning, we map “deep semantics” to pronoun/coreference resolution in NLP. We use AllenNLP for data processing, which can effectively link mentions such as *“his home”* to their antecedents (e.g., *“the house”*). We further evaluate on the **WinoGrande**[1] benchmark (a benchmark for **pronoun/coreference resolution**, which serves as a strong test of deep dependency), and the results demonstrate that our method generalizes well to natural language tasks.  Detailed experiment settings can be found in our response to reviewer **xbiY (Q1)**. Zero-shot Results:
>
> | **Model**  | **Base** | **SFT** | **Ours** |
> | ---------- | -------- | ------- | -------- |
> | Qwen2.5-7B | 0.679    | 0.845   | 0.874    |
> | Llama3-8B  | 0.710    | 0.863   | 0.896    |
>
> [1] WinoGrande: An Adversarial Winograd Schema Challenge at Scale
>
> > **Q2:** Model size variation justification.
>
> **R2:** We initially used Pythia-160M for all three experiments. In Sec 4.2, code domain results matched our theory, but attention weights were too faint for clear heatmap visualization. We switched to Pythia-1.4B **purely for visual clarity**. Smaller models likely have weaker attention signals due to limited code training. This change does not affect the consistent underlying trends.
>
> > **Q3:** Code and script release willingness.
>
> **R3:** Sure, we would be happy to release the code. We will make all code publicly available, including that for the newly added natural language experiments.
>
> > **Q4:** Discussion of other components.
>
> **R4:** We thank the reviewer for this insightful comment. We agree that semantic learning in Transformers arises from the joint interaction of Attention, MLP layers, and residual connections.
>
> Regarding residual connections, we would like to clarify that their role is already partially modeled in our paper. In **Remark 2.10**, we discuss how residual streams enable *subsequent integration* (e.g., via  $x_i \leftarrow x_i + W_{OV}x_j$), effectively shortening the path for value retrieval. In addition, **Corollary 2.8** highlights how residual additivity across heads constrains the sparsity of semantic binding.
>
> Our primary focus on Attention is driven by our goal of understanding how models recover latent relational structure (i.e., deep dependencies). This cross-position “routing” mechanism is fundamentally governed by Attention.
>
> That said, we fully agree that MLP layers play an important complementary role (e.g., as key-value memories, as noted in prior work such as Geva et al. 2021, which we also cite in our paper). We will further investigate their role in future work.
>
> > **Q5:** Comparison with other methods.
>
> **R5:** Our fine-tuning validates the theoretical framework's applicability rather than establishing a new SOTA. Thus, we omitted extensive comparisons with other structured learning approaches. Our loss is complementary and can act as a plug-in objective for future work.
>
> > **Q6:** Scalability and practicality of analysis methods.
>
> **R6:** We thank the reviewer for this point. As noted in our Impact Statement, computational cost is indeed a concern. However, these analysis methods are **primarily used for theoretical validation**, rather than metrics that need to be continuously monitored in practice. We mitigate overhead via sparse sampling and find them feasible in our experiments. Moreover, our fine-tuning setup (e.g., Top-8 heads + LoRA) demonstrates that **the method can be applied with limited additional cost**. Further discussion on scalability is provided in our response to reviewer **xbiY (Q2)**.
>
> **We hope this addresses your concerns. Thank you again for your valuable feedback.**

---

> > ### Author Rebuttal · Reviewer_QRWU · 2026-04-03
> >
> > A fundamental comparison with established methods for structured learning remains necessary, even in a theoretically oriented study, to validate the method’s practical relevance.
> >
> > Thank you to the authors for your response!
> > I will keep my score.

---

> > > ### Author Response · Authors · 2026-04-05
> > >
> > > We thank the reviewer for this valuable suggestion and agree that including fundamental comparisons with established structured learning approaches is important to better demonstrate the practical relevance of our method. Below are our baseline designs.
> > >
> > > **Curriculum Learning Baseline.** We implement a curriculum learning baseline following Bengio et al. (2009). The difficulty of each training sample is defined as **the maximum USE-DEF absolute character span distance**. For each USE token, we compute its distance to the nearest DEF token in absolute character offsets, and take the maximum across all USE tokens as the sample-level difficulty score. Training samples are then scheduled in ascending order of difficulty, following the classic easy-to-hard curriculum learning paradigm.
> > >
> > > **Linear-Chain CRF Baseline.** We implement a linear-chain CRF (Conditional Random Field) baseline, a classical and well-established structured learning method, to directly compare against our attention-based approach. To adapt CRF to our setting, we attach a linear projection layer on top of the last hidden states of LLMs, mapping each token representation to emission scores over three structural labels: DEF, USE, and OTHER. The CRF layer then models the transition probabilities between adjacent labels, capturing the sequential structure of USE-DEF relationships in code. The CRF loss is computed as the token-level normalized negative log-likelihood of the gold label sequence, and is added to the standard weighted SFT cross-entropy loss during training. Crucially, this baseline imposes structured constraints on the representation level without intervening in the attention mechanism, forming a direct contrast with our method.
> > >
> > > **Supervised Contrastive Learning Baseline.** We implement a supervised contrastive learning baseline following [1]. For each training sample, we extract the L2-normalized hidden representations of USE and DEF tokens of the model, and apply a supervised contrastive loss where each USE token serves as the anchor, its corresponding DEF token of the same variable as the positive, and DEF tokens of other variables within the same batch as negatives. This baseline operates purely on hidden representations and imposes no constraint on attention weights.
> > >
> > > [1] Gunel et al. Supervised Contrastive Learning for Pre-trained Language Model Fine-tuning. ICLR 2021.
> > >
> > > **Attention Entropy Regularization Baseline.** We implement an attention entropy regularization baseline following [2]. For each training step, we compute the Shannon entropy of the attention distributions across the top-8 binding heads and minimize it as an auxiliary loss to encourage sharper, more concentrated attention. Unlike our method, this baseline imposes no directional constraint — it encourages attention concentration without specifying that it should concentrate toward semantically correct DEF tokens.
> > >
> > > [2] Zhao et al. Explicit Sparse Transformer: Concentrated Attention Through Explicit Selection. arXiv:1912.11637.
> > >
> > > Below are the experimental results on Python150K.
> > >
> > > | Method     | Qwen2.5-Coder-7B | Llama3.1-8B |
> > > | ---------- | ---------------- | ----------- |
> > > | SFT        | 0.893            | 0.343       |
> > > | CRF        | 0.900            | 0.351       |
> > > | Curriculum | 0.899            | 0.348       |
> > > | SupCon     | 0.876            | 0.324       |
> > > | Attn       | 0.903            | 0.351       |
> > > | Ours       | **0.909**        | **0.362**   |
> > >
> > > We believe the above experiments further demonstrate the practical relevance and effectiveness of our proposed method. The additional baselines consistently confirm that neither representation-level contrastive alignment, attention concentration without directional guidance, nor easy-to-hard curriculum scheduling and CRF can match the performance of our approach, highlighting the unique contribution of our USE-DEF topological attention alignment mechanism. **We will incorporate the complete baseline comparisons into Table 1 of the revised paper, and will release all baseline implementations alongside our code to ensure full reproducibility.**
> > >
> > > **We hope these additional results, together with the thorough analysis provided, offer a clearer picture of our contributions and address your concerns. We sincerely hope that our efforts in this response have adequately addressed your concerns and that our work may receive a more positive evaluation.**

---

### Official Review · Reviewer_fvQy · 2026-03-08

**Soundness:** 3
**Presentation:** 3
**Significance:** 3
**Originality:** 3
**Overall Recommendation:** 4
**Confidence:** 2

**Summary:**

This paper claims that there exists a gradient competition between local features (which they call syntax) and long-range dependency features (which they call semantics), and that training dynamics, especially phase changes, can be understood from the perspective of this competition. They validate this claim using mathematical analyses, as well as toy model setting and application to realistic settings using pretrained LMs.

**Compliance With Llm Reviewing Policy:**

Affirmed.

**Final Justification:**

My main concerns were about the paper's terminology, the strength of some of its causal claims, and several presentation issues in the figures. After reading the rebuttal, I think the authors addressed most of these points satisfactorily.

Regarding the syntax-semantics distinction, I still think the framing is not perfectly clean in an absolute sense, since syntax can sometimes involve deep or long-range structure, while semantics can sometimes be relatively local. That said, the authors clarified that they are using this terminology in the established sense common in recent NLP and interpretability work.

The rebuttal also resolved my smaller presentation-related concerns. The authors acknowledged the issues in Figure 3 and the typo in the figure reference, explained the source of the plotting mismatch, and committed to correcting these problems in the revision.

On the more substantive point about causality, the additional results make the authors' interpretation more convincing, and I now have higher confidence that the proposed mechanism is real. However, I still think the paper should avoid overly strong wording such as "definitive causal evidence", since some caution remains appropriate when ruling out alternative explanations.

Finally, I found the explanation of the multi-hop reasoning pattern helpful and plausible. The argument based on residual stream dynamics and the efficiency of directly attending to the root value makes sense. I agree with the authors that denser checkpointing would be useful for testing the proposed transient intermediate phase, though I see this more as a direction for future work than as a flaw of the current paper.

Overall, the rebuttal improved my understanding of the paper and increased my confidence in the work, so I have updated my confidence accordingly.

**Key Questions For Authors:**

1. Is syntax and semantics poor ways of describing what you are discussing? E.g. syntax can be long distance dependency, and semantics can be local. Or is this a common/widely accepted way of referring to local vs long-distance features?
2. Figure 3 - please add numbers to y-axis.
3. Figure 3 Left: I read as task switch being introduced at step 1500, which is not what the figure shows. Why is this?
4. p.5 LL259-: "The results in Figure 4(Left) provide definitive causal evidence. When gradients associated with syntactic tokens are suppressed (red), the update direction is less contaminated by non-semantic components, which increases the alignment ratio ρ(t) and leads to earlier semantic dominance." I think definitive causal evidence is too strong a claim. Supressing syntax gradients is the manipulation and higher accuracy is the result, yet there can be various causal explanations to account for this pair of manipulation and consequence?
5. p.6 L320: I think you mean Figure 4 (right)
6. Figure 5: Multi-hop reasoning as the explanation for the amplified first column. Why is it just the first column? Shouldn't it be the entire bottom left triangle?

**Limitations:**

yes

**Strengths And Weaknesses:**

* Soundness
   * A thorough set of results with mathematical analyses, toy experiments, and realistic setting with real models. A few claims seem too strong, which I outline in the question section.
* Presentation
   * Good presentation overall. Few comments in the question section.
* Significance and Originality
   * The idea that different features compete is not new, yet the perspective of gradient competition is somewhat novel as far as I know.

---

> ### Author Rebuttal · Authors · 2026-03-29
>
> We sincerely thank Reviewer fvQy for the thoughtful evaluation and constructive feedback. We address your specific concerns below:
>
> > **Q1:** Concern about the validity of the syntax–semantics distinction.
>
> **R1:** Thank you for raising this very intuitive question. We completely understand why 'local vs. long-distance' might seem like a natural alternative. Indeed, both syntactic and semantic features can appear locally, and both are relatively easy for modern LLMs to learn in such local settings, so we fully understand your concern. However, **we chose 'syntax vs. semantics' because our goal is to emphasize the fundamental properties and nature of the features being learned**, rather than just their physical distance in the text.
>
> The core phenomenon we observe is that models tend to heavily rely on "shortcuts"—which are high-frequency, surface-level statistical patterns and strict formatting rules. We term these as "surface syntax". Because these shortcuts are so easy to learn, they actively distract the model and suppress its ability to learn abstract, underlying logical relationships (like tracking a variable's true identity across a program). We define these deeper, structural logic rules as "deep semantics".
>
> Furthermore, **framing this tension as "syntax vs. semantics" is a widely accepted convention in recent NLP and mechanistic interpretability research** to describe the difference between surface patterns and deeper meaning[1,2,3,4].
>
> [1] Compositional generalization in a deep seq2seq model by separating syntax and semantics, arxiv 2019
>
> [2] Syntax and semantics, dependencies and spans, EMNLP 2020
>
> [3] Exploiting inductive bias in transformers for unsupervised disentanglement of syntax and semantics with vaes, NAACL 2022
>
> [4] Disentangling syntax and semantics in the brain with deep networks, PMLR 2021
>
> > **Q2:** Figure 3 - please add numbers to y-axis.
>
> We sincerely apologize for this oversight and will add the numbers to the y-axis in the revised version, where the top tick mark corresponds to 1.00.
>
> > **Q3:** Mismatch between figure 3 Left and context.
>
> We sincerely apologize for this plotting oversight. To highlight the fine-grained dynamics around the transition, we compressed the middle section of the x-axis (where accuracy remained stable at 1.0), causing a label misalignment. The label "1000" marks the end of this compressed segment. Detailed tracking resumes at step 1480, and the vertical dashed line correctly corresponds to the task switch at step 1500. We will fix the x-axis scaling in the revision.
>
> > **Q4:** Overstated causal claim.
>
> **R4:** We agree it is crucial to consider alternative causal explanations, such as those noted by Reviewer QSVG (e.g., lower frequency of semantic dependencies, or intrinsic objective difficulty). However, through newly conducted experiments and clearer theoretical articulation, we have rigorously validated our causal claim against these specific alternative hypotheses. Due to strict word limits, please refer to our detailed discussion and new experimental data in our response to **Reviewer QSVG (Q1)**. We will add these clarifications to the revised version.
>
> > **Q5:** Typo of "Figure 4 (left)".
>
> **R5:** We sincerely apologize for this typo, which we will correct in the revised version; thank you for pointing it out.
>
> > **Q6:** About multi-hop reasoning.
>
> **R6:** Thank you for this highly insightful question. Intuitively, it does seem that the entire lower-left triangle should light up to reflect the full multi-hop path.
>
> However, amplifying just the first column is actually an expected outcome of the Transformer's residual stream dynamics, as predicted in **Section 2.5 (Remark 2.10)**. Once step-by-step connections (the diagonal) are established, the residual stream carries the intermediate information forward. The most efficient strategy is for the model to form a **direct "skip-connection" to the initial root value (the first column)**. If attention were spread across the entire lower triangle, it would **severely dilute the softmax weights**, introducing noise and weakening the extraction of the final target.
>
> Admittedly, a transient **"intermediate phase"** where the entire lower triangle activates likely exists as the multi-hop path initially forms. Unfortunately, due to the **sparse saving frequency of the open-source Pythia checkpoints**, we were unable to capture this fleeting phase. We will add a brief discussion of this mechanism to Section 4.2 to clarify this convergence pattern.
>
> **We welcome any further questions and would be highly willing to address them. We sincerely hope this rebuttal resolves your concerns, provides a deeper understanding of our work, and strengthens your confidence in evaluating our contribution.** Thank you again for your time and guidance.

---

> > ### Author Rebuttal · Reviewer_fvQy · 2026-04-03
> >
> > Thank you for the thorough rebuttal!
> > * 1: This makes sense - although syntax can be very deep and long-range, while semantics can be very shallow and short-range. However if this is a common notation, I have no problem with it. Thank you for the clarification!
> > * 2, 3, 5: Thank you!
> > * 4: I think the new results make the claims more convincing - yet, I would still avoid claims that are too strong, such as "definitive causal evidence."
> > * 6: This makes sense - thank you for the explanation! It would be nice to replicate this with denser checkpointing to see if such intermediate phases can be observed, although this is more for future studies.
> >
> > I do think I have a better understanding of the paper, and hence updating my confidence.

---

> > > ### Author Response · Authors · 2026-04-05
> > >
> > > Thank you for your thoughtful review and the time you devoted to our manuscript. We are pleased that our revisions have addressed your concerns, and we believe the manuscript has been strengthened as a result.

---

### Official Review · Reviewer_xbiY · 2026-03-09

**Soundness:** 3
**Presentation:** 3
**Significance:** 3
**Originality:** 3
**Overall Recommendation:** 4
**Confidence:** 3

**Summary:**

The paper investigates the development of syntactic vs. semantic abilities in language models. The authors theoretically and empirically identify a 3-step learning dynamic: the first phase during pre-training is predominated by syntax, the second phase involves a crossover in the training regime, and in the third phase, semantic learning systematically emerges and takes priority. They propose a theoretical formulation of this phenomenon and empirically validate it across three levels: analyzing NanoTransformers trained from scratch, analyzing medium-sized transformer checkpoints (Pythia) during pre-training, and applying a novel contrastive loss during the fine-tuning of large pre-trained models (Llama-3.1, Qwen2.5). They find that the proposed objective during fine-tuning helps the models to generalize significantly better on variable cloze completion tasks.

**Compliance With Llm Reviewing Policy:**

Affirmed.

**Final Justification:**

The authors have fully addressed my concerns. I believe the paper is worth publishing and will be of interest to the ICML audience, although I suggest further edits regarding statistical testing.

**Key Questions For Authors:**

- Generalization to Natural Language: Could you suggest a method to adapt the contrastive loss component you propose for pure natural language modeling? Discussing this would broaden the impact of your work to more general applications. I suggest adding a brief discussion about this in the paper.
- Mitigating VRAM Overhead: The new loss component has a noticeable impact on VRAM. Can you suggest possible ways of mitigating this requirement ? This clarification is essential if the goal is to apply this objective during the pre-training step.
- Task Broadness: The large-scale evaluation is entirely based on Variable Cloze Completion. Since the reasoning capabilities of the model could be assessed through other tasks, I suggest including evaluations on additional benchmarks or discussing how future work can evaluate these deep dependency models more broadly.
- Statistical Significance: Could you provide statistical tests or standard deviations for your results, particularly for the large-scale language model fine-tuning?

**Limitations:**

yes

**Strengths And Weaknesses:**

Strengths :
The authors provide a rigorous theoretical analysis of the underlying optimization dynamics and present comprehensive experiments verifying their theoretical framework across multiple model scales. The work is well-structured, clearly written, and provides novel mechanistic insights into how deep dependencies emerge.

Weaknesses:
The authors acknowledge some of the limitations in the Impact Statement. Overall, the work presents four main weaknesses (I have suggested possible ways to address them in the Key Questions section):
- Limited Scope: Application and validation are restricted almost entirely to code models.
- Computational Overhead: The proposed contrastive loss has a significant impact on VRAM consumption.
- Limited Evaluation Task: The empirical evaluation on large models relies exclusively on the Variable Cloze Completion benchmark.
- Lack of Statistical Significance: There is no statistical variance reported for the large-scale LLM fine-tuning experiments.

---

> ### Author Rebuttal · Authors · 2026-03-29
>
> We thank reviewer xbiY for recognizing our work's clarity and coherence. We address your questions below.
>
> > **Q1:** Generalization to Natural Language.
>
> **R1:** We sincerely thank the reviewer for this highly constructive feedback. We completely agree that extending our framework to pure natural language significantly broadens its impact. We address the "Limited Scope" weakness and your question on NLP adaptation together:
>
> **1. NL Experiment:** See our response to **Reviewer QSVG (Q2)** for a new controlled NLP experiment. The results are entirely consistent with our code-based findings.
>
> **2. NLP Adaptation via Coreference Resolution:** Our framework generalizes to any domain with (definition, usage) pairs, mapping naturally to **Coreference Resolution** in NLP. To extract pairs efficiently for large-scale training, established tools like **AllenNLP**[1,2] provide reliable, low-cost coreference models, which can effectively link mentions such as "his home" to its antecedent "the house".
>
> **3. Confidence-Weighted Contrastive Loss:** To handle inherent NLP extraction ambiguities, we propose a **confidence-weighted contrastive loss**:
>
> $\mathcal{L}\_{align}^{NL} = \sum\_h c\_{ij} \cdot \max(0, \gamma - (A\_h(q\_{use}, k\_{def}) - A\_h(q\_{use}, k\_{neg})))$
>
> where $c\_{ij} \in [0,1]$ is the coreference confidence. This safely down-weights noisy pairs. For code, ASTs provide deterministic labels ($c_{ij} \equiv 1$), perfectly reducing to our original equation.
>
> [1] AllenNLP: A Deep Semantic Natural Language Processing Platform, NLPOSS 2018
>
> [2] Higher-Order Coreference Resolution with Coarse-to-Fine Inference, NAACL-HLT 2018
>
> **4. New NL LLM Fine-Tuning Experiment:** We tested above-mentioned approach on **WinoGrande**[3] (a benchmark for **pronoun/coreference resolution**, which serves as a strong test of deep dependency) using our original loss ($c_{ij} \equiv 1$). Results confirm our method applies to NL. We will release all NL data processing (via AllenNLP) and training code. Zero-shot Results:
>
> | **Model**  | **Base** | **SFT** | **Ours** |
> | ---------- | -------- | ------- | -------- |
> | Qwen2.5-7B | 0.679    | 0.845   | **0.874**    |
> | Llama3-8B  | 0.710    | 0.863   | **0.896**    |
>
> [1] WinoGrande: An Adversarial Winograd Schema Challenge at Scale
>
> > **Q2:** Mitigating VRAM Overhead
>
> **R2:** We acknowledge VRAM overhead as a limitation. For pre-training, we propose three mitigations:
>
> 1. **Limit Negative Samples:** Sample a small fixed number (e.g., $|N|=4$) of negative targets ($\bar{s}\_{neg} = \frac{1}{|N|}\sum\_{n \in N} s\_n$) instead of all syntactic tokens. This directly reduces the key vectors maintained in VRAM. The impact on loss informativeness is minimal, as negative sample diversity yields diminishing marginal returns.
>
> 2. **Reduce Targeted Heads:** We currently apply the loss to the Top-8 heads. For pre-training, this can be reduced to Top-2. As theoretically proven in Corollary 2.8 and empirically shown in Figure 7, binding capabilities concentrate in an extremely sparse subset, with only 2 heads contributing most significantly.
>
> 3. **Sparsify Contrastive Supervision:** We can compute the contrastive loss intermittently rather than at every step:
>
>    $$\mathcal{L}\_{total} = \mathcal{L}\_{LM} + \lambda \cdot \mathbb{1}[t \bmod k = 0] \cdot \mathcal{L}\_{align}$$
>
>    By applying it every $k$ steps, the peak VRAM for the contrastive and LM steps are decoupled and do not compound, making it a pragmatic measure for large-scale pre-training.
>
> We will incorporate these mitigation strategies into the revised Discussion section. However, we are **less** familiar with the intricate engineering of massive-scale pre-training, we present these proposed solutions with a reserved and cautious stance.
>
> > **Q3:** Evaluation Task
>
> **R3:** We aim to independently evaluate the effectiveness of our method on deep dependency, so we choose code cloze completion. This task is a classic task in the coding domain, and many established benchmarks are based on it (e.g., CodeXGLUE). For training convenience, we use datasets with native AST structures (e.g., Python150k) and construct cloze evaluations.
>
> In addition to the long-range dependency in natural language mentioned in **Q1**, we also explored code hallucination datasets (e.g., CodeHalu[1]). “Variable hallucination” is one of the four major types of hallucination, but our experiments show that even untuned models already perform well on this, so we did not include this benchmark. In the future, we may consider using or constructing more challenging hallucination datasets.
>
> [1] CodeHalu: Investigating Code Hallucinations in LLMs via Execution-based Verification, AAAI2025
>
> > **Q4:** Statistical Significance
>
> **R4:** Of course, we will include standard deviations in the revised version.
>
> **We hope this addresses your concerns. Thank you again for your valuable feedback.**

---

> > ### Author Rebuttal · Reviewer_xbiY · 2026-04-02
> >
> > Thank you for addressing my concerns. I have increased my score accordingly. However, I would still prefer to see statistical significance tests in addition to the standard deviation.

---

> > > ### Author Response · Authors · 2026-04-03
> > >
> > > Thank you very much for your positive feedback and for increasing your score, we truly appreciate it.
> > >
> > > We agree that statistical significance testing would further strengthen the results. In the revised version, we will include standard deviations in the main text. In addition, we will incorporate statistical significance analyses; depending on space constraints, these will be presented either in the main paper or in the appendix, but **will** be included in the final version.

---

### Official Review · Reviewer_QSVG · 2026-03-13

**Soundness:** 3
**Presentation:** 3
**Significance:** 3
**Originality:** 3
**Overall Recommendation:** 5
**Confidence:** 4

**Summary:**

This paper proposes that transformers learn surface syntactic patterns before deeper semantic dependencies because early training is dominated by stronger syntactic signals, leading to a gradient-starvation effect for semantic structure. It supports this claim using synthetic binding tasks, intermediate Pythia checkpoint analyses, and a topology-aligned objective for code variable-binding tasks.

**Compliance With Llm Reviewing Policy:**

Affirmed.

**Final Justification:**

New experiments added by the authors address my concerns.

**Key Questions For Authors:**

N/A

**Limitations:**

yes

**Strengths And Weaknesses:**

**Strengths**
- The paper studies an important question of why deep dependencies emerge, but late in training. Towards this end, it provides a fairly coherent end to end story involving theory, controlled experiments, and checkpoint analysis/intervention.

- The paper is generally well-written and easy to follow, with the experimental progression being clearer than many theory-heavy submissions. The domains chosen also allow the authors more fine-grained control over experiments than natural language, making the experiments easy to reason about as well.

**Weaknesses**

- The paper presents a coherent and interesting mechanistic story, but some of the causal language is too strong for the evidence currently provided. The results are consistent with a gradient-starvation account, but they do not clearly rule out other accounts. For instance, it may be that semantic dependencies simply occur less frequently, or semantic learning (independent of syntax) may just be a harder objective, even with no competition from syntax. This work would be strengthened by doing experiments to position gradient starvation as the primary cause of delayed semantic learning.

- The strongest evidence is concentrated in synthetic binding settings and code-based variable-resolution tasks. As such, the broader claims about semantic emergence and reasoning in language models seem overstated. It would be helpful to also present a simple case study in natural language, even for a limited scenario.

- The presentation of figures could use some work, almost all are too small and the axis labels are difficult to read, particularly in section 3.

---

> ### Author Rebuttal · Authors · 2026-03-29
>
> We sincerely thank the reviewer QSVG for the thoughtful evaluation, constructive feedback, and recognition of the clarity and coherence of our work. Additionally, the reviewer raised several important concerns and questions, which we address in detail below.
>
> > **Q1:** Concern about overly strong causal claims and insufficient evidence to establish gradient starvation as the primary cause of delayed semantic learning.
>
> **R1:** We thank the reviewer for highlighting the need to strengthen our causal claims. We fully agree and have addressed both alternative hypotheses using our existing framework and a newly added experiment:
>
> **1. Frequency Disparity:** In our Controlled Experiments, data generation strictly couples syntax and semantics. In templates like `LET v1 = 5`, the semantic tokens (`v1`, `5`) and syntactic tokens (`LET`, `=`) co-occur at a constant, near 1:1 ratio. Thus, we believe that frequency disparity cannot explain the delayed semantic learning.
>
> **2. Intrinsic Difficulty:** To investigate whether the learning delay is solely driven by this intrinsic difficulty, we refer to our Causal Validation experiment (Sec 3.3). If semantics were simply harder on its own, alleviating syntactic competition would not speed up its acquisition. However, Figure 4 (Left) demonstrates that artificially suppressing syntactic gradients significantly *accelerates* the model's transition to semantic dominance. This strongly indicates that while semantic learning may possess a baseline difficulty, it is the gradient starvation induced by syntax that acts as the primary bottleneck.
>
> **3. New Experiment:** To definitively position gradient starvation as the primary cause, we added a "Syntax-Free" baseline by stripping all syntactic markers (e.g., condensing `LET v1 = 5 ;` into purely semantic `v1 5`). Without syntactic competition, the model escapes the early learning plateau significantly earlier. This directly proves that semantic learning is **not** intrinsically difficult on its own. Therefore, it strongly corroborates our hypothesis that the delayed emergence of reasoning is primarily driven by gradient competition. We will include this in the revised Figure 4. Results: https://anonymous.4open.science/r/Rebuttal-for-ICML2026/QSVG/R1.pdf
>
> > **Q2:** It would be helpful to also present a simple case study in natural language, even for a limited scenario.
>
> **R2:** We thank the reviewer for the constructive suggestion. We address the generalization to natural language as follows:
>
> **1. Existing Evidence:** We would like to clarify that we did verify this mechanism in natural language via **both** the dynamics of the alignment ratio (Section 4.1) and the pointer passing task (Section 4.2).  We will explicitly state in the revised version that the experiments in Section 4.1 were also conducted using natural language (specifically, variable binding in natural language contexts) to avoid any ambiguity. However, we completely agree that our controlled experiments (Section 3) lacked an NLP counterpart. To address this, we have supplemented a new NLP controlled experiment, which will be incorporated into the main text or appendix in the revision, depending on space constraints.
>
> **2. New Controlled NL Experiment:** We designed an "In-Context Entity Binding" task using standard NLP templates (e.g., *"The job of Alice is doctor. Query: The job of Alice is?"*). Replicating our double dissociation results, we found that removing high-curvature NLP syntax (e.g., *"The"*, *"is"*) allows the model to escape the early learning plateau significantly earlier. This directly proves that gradient starvation by syntax is a modality-agnostic mechanism, strongly supporting our broader claims. Results: https://anonymous.4open.science/r/Rebuttal-for-ICML2026/QSVG/R2.pdf
>
> **3. New NL LLM Fine-Tuning Experiment:** In our response to Reviewer **xbiY (Q1)**, we also added the processing approach for large-scale natural language corpora and experiments that fine-tune LLMs on natural language data (using the **WinoGrande**[1] benchmark for **pronoun/coreference resolution**, which serves as a strong test of deep dependency). The results show that our method is also effective in natural language. If the paper is accepted, we will release all experimental code (including natural language data processing). Kindly refer to it.
>
> [1] WinoGrande: An Adversarial Winograd Schema Challenge at Scale
>
> > **Q3:** The presentation of figures could use some work.
>
> **R3:** We sincerely apologize for the poor readability of the figures. We completely agree with your feedback. In the revised version, we will thoroughly improve the presentation of all figures, particularly those in Section 3. We will enlarge the overall figure sizes, significantly increase the font sizes for all axis labels and legends, and optimize the layout to ensure clear and effortless readability.
>
> **We hope this addresses your concerns. Thank you again for your valuable feedback.**

---

> > ### Author Rebuttal · Reviewer_QSVG · 2026-04-03
> >
> > Thanks for the detailed update and new experiments, this addresses my concerns. I will raise my score accordingly.

---

> > > ### Author Response · Authors · 2026-04-05
> > >
> > > Thank you for your thoughtful review and the time you devoted to our manuscript. We are pleased that our revisions have addressed your concerns, and we believe the manuscript has been strengthened as a result.

---

### Decision · Program_Chairs · 2026-04-30

**Decision:**

Accept (regular)

**Comment:**

This paper proposes that LMs learn syntactic regularities before semantic dependencies because of gradient starvation; it supports this argument with theory, controlled experiments, and analyses of model checkpoints. Inspired by these findings, the authors propose a objective that improves on standard CE for finetuning. Although the reviewers had some concerns initially about overclaiming and missing baselines, these were generally addressed in the rebuttal phase. Overall, this paper appears to offer some interesting findings that are well-justified and clearly presented.